# Carbon budgets for an irrigated intensively-grazed dairy pasture and an unirrigated winter-grazed pasture

John E. Hunt[1], Johannes Laubach[1], Matti Barthel[1,2], Anitra Fraser[1], and Rebecca L. Phillips[1]

[1]Landcare Research, P.O. Box 69040, Lincoln 7640, New Zealand
[2]Department of Environmental Systems Science, ETH Zürich, 8092 Zürich, Switzerland

*Correspondence to*: Johannes Laubach (laubachj@landcareresearch.co.nz)

**Abstract.** Intensification of pastoral agriculture is occurring rapidly across New Zealand, including increasing use of irrigation and fertiliser application in some regions. While this enables greater gross primary production (GPP) and livestock grazing intensity, the consequences for the net ecosystem carbon budget (NECB) of the pastures are poorly known. Here, we determined the NECB over one year for an irrigated, fertilised, and rotationally-grazed dairy pasture and a neighbouring unirrigated, unfertilised, winter-grazed pasture. Primary terms in the NECB calculation were: net ecosystem production (NEP), biomass-carbon removed by grazing cows, and carbon (C) input from their excreta. Annual NEP was measured using the eddy-covariance method. Carbon removal was estimated with plate-meter measurements calibrated against biomass collections, pre- and post-grazing. Excreta deposition was calculated from animal feed intake. The intensively-managed pasture gained C (NECB = 103 ±42 g C m$^{-2}$ yr$^{-1}$) but would have been subject to a non-significant C loss if cattle excreta had not been returned to the pasture. The unirrigated pasture was C-neutral (NECB = −13 ±23 g C m$^{-2}$ yr$^{-1}$). While annual GPP of the former was almost twice that of the latter (2679 vs. 1372 g C m$^{-2}$ yr$^{-1}$), ecosystem respiration differed by only 68 % between the two pastures (2271 vs. 1352 g C m$^{-2}$ yr$^{-1}$). The ratio of GPP to the total annual water input of the irrigated pasture was 37 % greater than that of the unirrigated pasture, i.e. the former used the water input more efficiently than the latter to produce biomass. The NECB results agree qualitatively with those from many other eddy-covariance studies of grazed grasslands, but they seem to be at odds with long-term carbon-stock studies of other New Zealand pastures.

## 1 Introduction

Current and predicted trends in global agriculture include that grazed grasslands are increasingly managed more intensively, through application of irrigation and fertilisers (Thornton, 2010). This aims to significantly enhance gross primary production (GPP) of pasture and thereby support more frequent rotational grazing, at higher animal densities (Tilman et al., 2001). There are questions regarding the economic and environmental sustainability of this kind of intensification, particularly if water supplies needed for irrigation become limited. Also, the effects of irrigation and fertiliser application on the carbon (C) budget are not clear. Greater GPP may lead to greater soil organic carbon (SOC) stocks, but for pasture, the

transfer of atmospheric C to the SOC pool is largely dependent on grazing and irrigation management decisions (Ammann et al., 2007; Merbold et al., 2014). For some pastures, gains in GPP were offset by ecosystem respiration (ER) and C export (Ammann et al., 2007; Kelliher et al., 2012), so that a net loss of C and a negative net ecosystem carbon balance (NECB) resulted. For others, C inputs exceeded losses, leading to a positive NECB (Ammann et al., 2007; Merbold et al., 2014).

Grasslands cover 40 % of the land area of New Zealand and were historically used to support year-round sheep and beef pastoral agriculture (MacLeod and Moller, 2006). Conversion of sheep and beef pasture to dairy farming in the last two decades has placed New Zealand as the world's largest exporter of milk products (FAO, 2014). Initially, this conversion occurred mostly on large flat lowland areas with high rainfall, leading to a 37 % increase in the national dairy herd from 1994 to 2004 (Clark et al., 2007). Remaining, less productive land is now being targeted for intensive dairy farming by
means of pasture renewal, irrigation and fertiliser applications. These measures support the common New Zealand practice of high-density rotational grazing, where each pasture area is grazed several times per year for short periods of time ($1 - 2$ d) at an animal density of order 100 cows ha$^{-1}$ and then allowed to regrow undisturbed for typically $20 - 30$ d (Moir et al., 2010). The 60 % increase in milk-solids production between 1995 and 2005 in New Zealand (DairyNZ, 2014) is a combined result of the expanded area of intensively-managed pasture, increasing animal densities and the greater use of supplementary
feed (MacLeod and Moller, 2006; Ho et al., 2013). High-density grazing management results in repeated, sudden reductions of leaf area, in contrast to extensive grazing systems, where animals may graze the same pasture for weeks at a time and the leaf area is reduced much more gradually. Hence, intensive dairy-pasture management strongly differs from extensive management in that all the main factors which typically limit plant growth (water, nutrients and leaf area) are manipulated in a very controlled fashion, in order to maximise GPP.

Carbon stocks beneath grasslands in New Zealand are significant, representing over 50 % of the national soil C inventory (Tate et al., 2005). As the trend towards more intensive agriculture continues, it is central to New Zealand's economic and environmental sustainability to understand how intensive pasture management affects the ecosystem C balance. Historically, methods of assessing effects of management on SOC have been largely based on time series analyses of soil core data. Changes in belowground SOC "stocks" over time represent the net C sequestered (Stockmann et al., 2013; Phillips et al.,
2015). These types of studies require a large number of samples and time intervals that exceed five years (Ammann et al., 2007). Soil core samples taken at sites that had originally been sampled 20 to 40 years earlier indicate that intensive pasture management may reduce SOC stocks for some soil types in New Zealand (Schipper et al., 2014). Further, a sheep pasture that had been under irrigation for 60 years had lower SOC than unirrigated pasture on the same farm, possibly because additional ER of the irrigated pasture outpaced the additional C inputs (Kelliher et al., 2012).

Such long-term SOC studies do not establish direct links between net C uptake or loss and specific management activities (e.g. stocking and fertiliser-application rates, irrigation amounts, etc.) because of the coarse temporal resolution of the SOC sampling. Finer temporal resolution of GPP and ER in combination with detailed management data could be used to investigate how management interacts with climatic variables to alter the fundamental processes that drive C sequestration (Rajan et al., 2013). A possible approach that provides this finer temporal resolution is the eddy-covariance (EC) method

(Aubinet et al., 2000), which is widely used to construct the annual net ecosystem exchange (NEE) of $CO_2$ of various ecosystems from continuous measurements at sub-daily time scales. By convention, negative values for NEE represent C uptake by the ecosystem, and −NEE is equal to net ecosystem production (NEP) provided that gains or losses of dissolved inorganic C are negligible (Chapin et al., 2006). The EC method allows the partitioning of NEP into GPP and ER. It also

provides simultaneous measurements of evapotranspiration (ET), with which a link between the C and water budget of the ecosystem can be established. In order to derive the NECB of the ecosystem, the NEP obtained from EC measurements must be combined with data on other C imports and exports (Sect. 2).

So far, there is only one EC-based dataset of the NEP and NECB of an intensively-managed dairy pasture in New Zealand (Rutledge et al., 2015). In contrast to the SOC stock studies, this does not show a loss of C under the pasture. The same was

found for the majority of published EC-based studies on grazed grasslands elsewhere (Soussana et al., 2010; Rutledge et al., 2015). However, none of these included irrigation as a key management practice. Here, our main objective is to provide the first annual NEP and NECB of an irrigated dairy pasture in New Zealand. We also report the NEP and NECB of an adjacent unirrigated (and only weakly managed) pasture on the same farm. The NEP and NECB results are subjected to careful uncertainty analysis. We further aim to identify in which ways the management practices influenced both NEP and NECB;

to this end, we analyse the budgets and temporal variations of GPP, ER, water inputs and ET for both pastures.

## 2 The carbon budget of a pasture system

Generalising the notation of Chapin et al. (2006), the NECB and NEP of an ecosystem are related by the equation

$$NECB = NEP + F_{import} - F_{export} \qquad (1)$$

where NEP is the balance of $CO_2$ uptake and $CO_2$ emissions:

$$NEP = GPP - ER \qquad (2)$$

while $F_{import}$ and $F_{export}$ represent the non-$CO_2$ gains and losses of carbon for the ecosystem, respectively. These need to be specified dependent on the characteristics of the ecosystem.

For a pastoral system, it is necessary to first decide whether the grazing animals are considered as internal parts of the

system, or as agents of import and export (Felber et al., 2015). The former approach is suitable where grazing is extensive and grazing periods stretch over several weeks (e.g. Soussana et al., 2007), or where insufficient information is available on the animals' locations and movements over time (e.g. Mudge et al., 2011). Here, we employ the alternative approach because the animals' presence at the measurement sites was well-recorded and amounted to only about 3 % of the year. The grazing

cows are treated as agents of export which quickly removed large amounts of biomass, expressed by a term $F_{grazing}$. The cows were also agents of C import by depositing dung and urine on the pasture, denoted $F_{excreta}$.

Other C exchanges arise from management practices or natural processes. The former include fertiliser application, $F_{fert}$, and exports via biomass harvest, which did not occur in the pasture system investigated here. A natural process relevant to pasture systems is the loss of dissolved organic carbon (DOC) via leaching, $F_{DOC}$, and another one is the net flux of methane ($CH_4$) from soil and excreta to the atmosphere, $F_{CH4}$ (negative if uptake dominates). The C budget considered in the following is thus written as:

$$NECB = NEP + F_{excreta} + F_{fert} - F_{grazing} - F_{DOC} - F_{CH4} \tag{3}$$

Treating the grazers as external to the pasture ecosystem means that periods when they were present must be excluded from the recorded NEP data. This is because animal respiration during intensive grazing events constitutes a major contribution to the measured $CO_2$ fluxes, which then do not correctly represent the $CO_2$ exchange between the pasture-soil system and the atmosphere. The C respired by the animals originates from the biomass ingested and is thus already counted in the C budget as an export term. Hence, $CO_2$ exchange data during grazing events were excluded and replaced with modelled estimates of NEP using an established gap-filling method. Like the respired $CO_2$, the C emitted as $CH_4$ by the cows also originates from the grazed biomass and is therefore not included in Eq. (3), and the same is true for the C exported in milk.

## 3 Methods

### 3.1 Site description

The study was located on a commercial dairy farm in Canterbury, New Zealand (43.593° S, 171.929° E, 204 m above sea level, slope 0.8° ±0.4°). Most of the farm was converted from dryland sheep production to dairy in 2008/09. Its pastures were then cultivated and re-seeded with ryegrass (*Lolium perenne* L.) and clover (*Trifolium repens* L.). A centre-pivot irrigation system (890 m length) was installed, and the circular area under the irrigator was divided into 19 equally sized sectors (13.8 ± 0.2 ha). One of our measurement sites was located in one of these sectors (Fig. 1). We denote this fenced pasture by "IFR" (irrigated, fertilised, rotationally grazed); for future reference in the Fluxnet network, the measurement site there has been named "NZ-BFm" (D. Papale, pers. comm., 2016). Adjacent to the irrigated area is an unirrigated and more extensively managed paddock, which we denote by "UUW" (unirrigated, unfertilised and winter-grazed). There, we installed the second of our measurement sites, whose future Fluxnet name will be "NZ-BFu". The measurements reported here are for the first year of operation at both sites, beginning in late winter (17 August) 2012 and including one full milking season (September 2012 to May 2013).

Intensive management of the IFR pasture included synthetic fertiliser application and dairy cows grazing multiple times from September to May (Fig. 2). Irrigation was applied approximately every 3 d in summer to maintain the soil's volumetric water content (VWC) above 0.2 $m^3$ $m^{-3}$. Grazing events were short and intensive (>100 cows $ha^{-1}$), typically lasting 1 to 2 d (Fig. 2). During a grazing event, a herd of about 430 cows (half the farm's total number) was restricted by temporary fencing

so that it completely surrounded the instrument site, ensuring that biomass removal was quick and even in all directions. The UUW pasture was not fertilised or irrigated and was grazed only once, just after the end of the milking season (Fig. 2).

The soil is a Lismore silty loam of recent, alluvial origin (Hewitt, 2010), and classified as Typic Dystrustepts (SSDS, 1993). An electromagnetic survey across the northern half of the IFR-pasture circle and the UUW pasture (combined area 157 ha) gave a coefficient of variation of of the soil's electrical conductivity of only 2.63 %, indicating a very homogeneous soil

(AgriOptics NZ Ltd., unpublished customer report, 2012). Near the surface (0 – 100 mm), the soil particle size distribution was 13 % sand, 58 % silt and 29 % clay, with 12 % stone content. In this layer, under the IFR pasture, C content and carbon/nitrogen ratio were 2.48 kg C $m^{-3}$ and 10.75, respectively, and pH was 5.7 (Sam Carrick, unpublished data, 2013).

## 3.2 Measurements of meteorological and soil variables

The same kinds of sensors were deployed at each of the two sites to provide relevant environmental data. Measured variables included air temperature and relative humidity (HMP-45A, Vaisala, Helsinki, Finland) at 1.25 m above ground, wind direction (W200P, Vector Instruments, Rhyl, UK), soil temperature at 20 mm depth (TCAV, Campbell Scientific, Logan, UT, USA), and triplicate profiles of soil temperature (copper/constantan thermocouples) and moisture (SM300, Delta-T Devices Ltd, Burwell, Cambridge, UK) at 50, 100, 250 and 500 mm depth. Precipitation at the UUW site and total water

input at the IFR site were recorded using tipping-bucket rain gauges (TB3, Hydrological Services, Warwick Farm, NSW, Australia); irrigation events and amounts were identified by comparing the water input received at both sites. Radiation measurements, at 1.6 m above ground, included photosynthetically-active photon flux density (SQ-120, Apogee Instruments, Logan, UT, USA), net radiation (R01, Hukseflux, Delft, The Netherlands), and the Normalised Difference Vegetation Index, NDVI (SKR 1800, Skye Instruments, Llandrindod Wells, UK). Sensor readings were recorded every 3 s by a datalogger

(CR3000, Campbell Scientific, Logan, UT, USA) and stored as half-hourly values.

## 3.3 Carbon dioxide and water vapour flux measurements

Identical eddy-covariance (EC) systems were deployed in 2012 in the IFR and UUW paddocks (Fig. 1) to measure fluxes of $CO_2$, $H_2O$, sensible heat and momentum. Each EC system consisted of a three-axis ultrasonic anemometer (Windmaster Pro,

Gill Instruments, Lymington, UK) and a compact closed-path ("enclosed") infra-red $CO_2/H_2O$ gas analyser (LI-7200, LI-COR Biosciences, Lincoln, NE, USA). When one of the sonic anemometers developed a transducer problem, requiring off-site repair by the manufacturer, a loan instrument (Windmaster, Gill Instruments, Lymington, UK) was employed at the

UUW site from 15 Feb to 24 Apr 2013. On 26 Jul 2013, the UUW sonic was replaced by another model (R3-50-1, Gill Instruments, Lymington, UK). The sensor geometry and mode of operation of all three sonic types are very similar, ensuring continuity of the turbulent flow measurements.

The heights of the IFR and UUW EC systems (centre of sonic path) were 1.86 m above the ground. For each site, the fetch
was homogeneous for at least 130 m in all directions, ensuring that a large fraction of the flux footprint was representative of the target paddock. Care was taken to minimise high-frequency attenuation (Lenschow and Raupach, 1991; Moore, 1986) with the use of a small measurement cell, short tube length (1.0 m) and high flow rate (12 L min$^{-1}$). To minimise flux loss due to sensor separation, the intake tube was attached underneath the sensing volume of the sonic anemometer (Kristensen et al., 1997). Initially, the tube was made of 3/8" Dekabon, with an inner diameter of 5.8 mm. On 27 Mar 2013, the tubes at
both sites were replaced with 1/4" Synflex tubing (3.9 mm inner diameter), to reduce both the transit time of the air and the inner tube surface area where sorption effects cause attenuation of the water vapour signal (Ibrom et al., 2007; Nordbo et al., 2013). The transit time from the intake to the measurement cell was 0.15 s originally, and 0.07 s after the tube change.

Raw EC data were sampled at 20 Hz and stored by an Analyzer Interface Unit (LI-7550, LI-COR Biosciences, Lincoln, NE, USA). The data were post-processed to compute half-hourly fluxes with EddyPro (Version 5.0, www.licor.com/eddypro).
Raw EC data were despiked and quality-tested (Vickers and Mahrt, 1997). Instantaneous wind vectors were corrected (Nakai and Shimoyama, 2012), then double-rotated and expressed relative to the mean streamlines. The $CO_2$ and $H_2O$ concentration data were shifted in time to remove their lag relative to the sonic anemometer data. This lag was the sum of the transit time given above, the time to exchange the gas analyser's cell volume of 0.016 L (0.08 s), and a fixed processing delay of 0.13 s (see LI-7550 manual). The concentrations were then converted to mixing ratios using temperature and pressure (Burba et al.,
2012) and spectrally analysed. Spectral corrections were applied, including those for low-frequency error (Moncrieff et al., 2004), vertical separation (Horst and Lenschow, 2009), and high-frequency losses (Fratini et al., 2012). The EddyPro software was modified to correct for an error in the application of the filter function[1]. The half-power frequency for high-frequency losses was usually 1.15 ($\pm$ 0.15) Hz for $CO_2$, and was humidity-dependent for water vapour, ranging from 0.05 Hz at relative humidity > 85 % to 0.8 Hz at relative humidity < 25 %.
Also obtained with EddyPro were estimates of the change of $CO_2$ storage from one averaging period to the next in the air layer beneath the EC system. Provided that advective fluxes are negligible, the sum of the EC flux (which represents the flux at measurement height) and the storage change is equal to the net $CO_2$ flux at the vegetation surface (Baldocchi, 2003).

---

[1] We are grateful to Gerardo Fratini (LI-COR Biosciences Inc.) for providing the modified software on our request and for the excellent discussion around this issue. The original EddyPro software (up to Version 5.1) applies the square root of the filter function, $H_{IIR}$, to the heat flux cospectrum, in order to estimate the cospectral loss, as expressed in Eq. (3) of Fratini et al. (2012). The cospectrum needs to be multiplied with $H_{IIR}$ itself and so using the square root of $H_{IIR}$ is wrong, as was already noted by Horst (1997). In other words, the cospectrum is effectively subjected to the same filter shape as the power spectrum of the filtered series. The filter used on one time series suppresses so much correlation that it matters little whether the same filter is applied to the other time series, as well. The impact of the correction is thus increased. LI-COR Biosciences have released the corrected application of $H_{IIR}$ in EddyPro Version 5.2 (G. Fratini, pers. comm.).

As an indirect quality check on the EC flux measurements, we assessed the energy balance closure at the two sites, and found it in line with results from other EC sites around the world (Appendix A).

### 3.4 Filtering and gap-filling of $CO_2$ flux data

### 3.4.1 Pre-filtering criteria

Dubious $CO_2$ flux data were removed with the following sequence of criteria: 1) periods of gas analyser maintenance or calibration, as indicated by a flow rate < 10 L min$^{-1}$, and periods of known sonic anemometer malfunction, 2) grazing events (for reasons given in Sect. 2), 3) periods of instationary flow and very low turbulence. The most effective practical criteria for 3) were a combination of a filter for the half-hour standard deviation of wind direction ($\sigma_{dir}$) and a filter for the ratio of

mean wind speed to friction velocity ($\bar{u}/u_*$). Half-hour runs with $\sigma_{dir} > 40°$ were excluded as instationary, since they often indicated sudden air mass changes leading to implausible outliers in one or more of the turbulent flux variables ($CO_2$, $H_2O$, sensible heat). Runs with $\sigma_{dir} < 1°$ were excluded as they indicated a lack of wind, and runs with $\bar{u}/u_* < 6$ were also excluded, with this condition potentially caused either by calm or a transition from one wind regime to another.

### 3.4.2 Moving-point threshold detection

After pre-filtering, the $CO_2$ flux from EC and the change in $CO_2$ storage below the measurement system were added, to obtain the net $CO_2$ exchange between pasture and atmosphere. The dataset was then separated into day- and night-time runs. The $CO_2$ surface fluxes at night were subjected to a moving-point-threshold (MPT) test, similar to that of Gu et al. (2005). The purpose of this test was to determine whether undetectable advection below the measurement height in the stable

nocturnal surface layer would lead to biased flux estimates, as has been found at a majority of EC sites (Gu et al., 2005; Barr et al., 2013). This procedure was performed twice for each site, once with $u_*$ as the sorting parameter, as is common practice, and once with the standard deviation of vertical wind ($\sigma_w$), as suggested by Acevedo et al. (2009), because $\sigma_w$ is a more direct indicator of turbulence intensity and less likely to be contaminated by low-frequency motion in the boundary layer. For the MPT test, the data were separated into temperature classes of 3 K width. Further classification, by either season or vegetation

height, only added scatter and was abandoned. An example of the results, for one temperature class, is shown in Fig. 3. With both turbulence parameters, the $CO_2$ flux (ER) increases at weak turbulence, as expected, and reaches a plateau at stronger turbulence (within statistical uncertainty). At both sites, the increase at weak turbulence is steeper when using $\sigma_w$, and at the UUW site, the data distribution in the region of increase is rather scattered when using $u_*$. Such qualitative differences were found for most temperature classes, which supports Acevedo et al. (2009) in that $\sigma_w$ should be the preferred filtering variable.

For both sites, the best overall threshold value was obtained either as $\sigma_w = 0.12$ m s$^{-1}$ (Fig. 3 a,c) or as $u_* = 0.15$ m s$^{-1}$ (Fig. 3 b,d). Due to the generally steeper and tidier rise of ER with $\sigma_w$ than with $u_*$, the $\sigma_w$ threshold was more selective: it

excluded only 41 % and 37 % of the pre-filtered night-time runs at the IFR and UUW site, respectively, while the $u_*$ threshold excluded 53 % and 51 %, respectively. Therefore we discarded $CO_2$ fluxes from runs with $\sigma_w < 0.12$ m s$^{-1}$. After that, the total fraction of data gaps (day + night) was 39.8 % at the IFR and 41.5 % at the UUW site. The missing flux values were gap-filled as described next.

### 3.4.3 Gap-filling and NEP partitioning

For gap-filling of EC fluxes and partitioning of NEP into GPP and ER, Eq. (2), we used the self-organising linear-output (SOLO) model (Hsu et al., 2002), which is an artificial-neural-network model. Given the repeated abrupt changes of the IFR's vegetation status due to grazing events, where the grazing event itself always gave rise to a gap period, the neural-

network approach was expected to perform better than process-based gap-filling models (where fluxes vary mainly in response to weather variables and the phenology is assumed to change only slowly). The SOLO algorithm was executed within the OzFluxQC software package (Version 2.9.5, available at https://github.com/OzFlux/OzFluxQC).

For each variable to be gap-filled, the user must specify the driver variables (regressors). Both $CO_2$ and $H_2O$ fluxes were expected to depend on the vegetation status, which we represented by NDVI as a driver variable. As the meteorological

drivers for $H_2O$ flux (ET) we selected available energy (net radiation minus soil heat flux), saturation humidity deficit, wind speed and air temperature. As further drivers for $CO_2$ flux (NEP), we selected photosynthetically-active photon flux density, soil temperature at 20 mm depth, and saturation humidity deficit. A gap-filled respiration flux (ER) was constructed by applying the SOLO algorithm to the filtered night-time $CO_2$ fluxes, with NDVI, soil temperature, and soil VWC at 50 mm depth as driver variables. GPP was obtained as the sum of NEP and ER, see Eq. (2).

The uncertainty of the annual means of NEP, GPP and ER to the procedures of low-turbulence filtering and gap-filling was assessed by repeating the gap-filling procedure with alternative values for the $\sigma_w$ threshold, of 0.08, 0.10 and 0.15 m s$^{-1}$ and also with three choices for conventional $u_*$ thresholds, of 0.12, 0.15 and 0.18 m s$^{-1}$. Varying the threshold in this way implicitly varies the fraction of gaps in the dataset, and so provides an indication how the annual mean fluxes depend on data availability. For one threshold choice, we also compared the gap-filled NEP to that computed with the process-based gap-

filling method of Barr et al. (2004). In this method, respiration during gaps is modelled as a logistic function of soil temperature at a shallow depth, photosynthesis is modelled as a function of incoming photon flux density using a Michaelis-Menten type equation, and in each model the linear factor in the numerator is allowed to vary slowly over time.

### 3.4.4 Footprint estimation and corrections

To assess the contributions to the EC fluxes from areas farther upwind than the target pasture, we used a spatially explicit footprint tool (Neftel et al., 2008). The computed footprint fractions were used on a run-by-run basis to apply a "footprint correction". Both the footprint fractions and the resulting flux corrections were analysed statistically, separately by season

and wind direction. At each site, the uncorrected nocturnal and daytime fluxes were summed separately and compared to their footprint-corrected counterparts. Details of the procedure and results are given in Appendix B.

### 3.5 Determination of plant dry matter removed by grazing

The standing biomass of the IFR grass canopy was determined before and after each grazing event from measurements of compressed canopy height made with a rising-plate meter (F200, FarmWorks, Feilding, New Zealand). The average height was determined from a total of 40 readings taken on transects along each of the cardinal directions from the instrument site. Each stored reading was an average of 10 measurements collected at about 1.5 m intervals.

Calibration of the plate meter was carried out prior to six of the grazing events. Sixteen 0.25 m$^2$ plots were selected to
represent the full range of biomass, from sparse to dense vegetation cover. The compressed canopy height was determined by the plate meter before the grass was cut at about 50 mm above ground (the observed mean grass height after grazing events). The collected biomass was oven-dried at 60 °C and the dry-matter (DM) weight determined. The plate meter's calibration was obtained by linear regression against the DM data and applied to the readings collected from across the paddock.

The average leaf area index was determined with a leaf area meter (LI-3100, LI-COR Biosciences, Lincoln, NE, USA) before and after a single grazing event in December, using biomass collected from 40 randomly located plots.

### 3.6 Estimation of other C imports and exports

The amount of faecal DM deposited by the cows was estimated as a fixed fraction (1 − digestibility) of the mass of DM
ingested (the sum of grazed and supplementary feed). The volume of urine deposited was estimated with typical values from the literature, and the C contents of dung and urine were based on values measured elsewhere in New Zealand (see Appendix C for details). A small fraction of the excreta was deposited outside the grazed pasture, during milking and in transit to and from the milking shed. The excreta deposited on the milking platform were collected into an effluent pond and later returned to the pasture, and hence gaseous emissions during effluent storage are the only process causing a minor loss
of C from this fraction. Regarding the excreta deposited in transit, the major part still fell onto pasture areas within the irrigated circle. Only the excreta deposited onto farm tracks were not returned to the pasture, but the time fraction spent by the cows on these tracks was minor and is ignored here.

Detailed fertiliser input records for both paddocks were obtained from the supplying company. The amount of C added by fertiliser application was calculated from the application rate and the chemical composition of the fertiliser.
Export of C as DOC ($F_{DOC}$) was not measured, so this was estimated using measurements of leachate collected from similar soil (S. M. Thomas, pers. comm., 2015). The net exchange of $CH_4$ of the pasture including excreta deposited onto it ($F_{CH4}$) was measured by a combination of two micrometeorological methods (Laubach et al., 2015, this issue).

# 4 Results

## 4.1 Environmental conditions

Mean annual air temperatures at the UUW (11.1 °C) and IFR (10.9 °C) sites were close to the 30-year mean of 10.9 °C
recorded nearby at Horarata by the National Institute of Water and Atmospheric Research (cliflow.niwa.co.nz). During
summer, soil temperature of the UUW pasture tended to be higher than that of the IFR pasture (Fig. 4a). Soil VWC at 50 mm
depth ranged from 0.06 to 0.5 $m^3$ $m^{-3}$ for the UUW pasture and from 0.25 to 0.6 $m^3$ $m^{-3}$ for the IFR pasture. The VWCs for
both pastures were similar until late spring (November) but diverged in summer, and VWC of the UUW pasture remained
lower than that of the IFR pasture for the remainder of the year (Fig. 4b). Total annual rainfall was 1014 mm, which
exceeded the 30-year average of 839 mm (cliflow.niwa.co.nz). However, this included 273 mm that fell in an extremely wet
winter period in June 2013 (Fig. 4c), with no effect on the grazing season which had already finished. During the warmer
half of the year (Oct – Mar) the total rainfall was 407 mm, 6 % less than the long-term mean of 431 mm. This amount of
summer rain was not sufficient for the needs of pasture vegetation. Therefore, in the IFR paddock it was supplemented by
425 mm of irrigation.

For the IFR pasture, repeated sudden NDVI reductions are evident (Fig. 4d), indicating biomass removal by grazing.
Following each grazing event, NDVI steadily increased to pre-grazing values ($\geq 0.8$) in less than a month. For the UUW
pasture, NDVI remained undisturbed at values $> 0.8$ from October until mid-December and then declined steadily
throughout the long dry period until mid-March. From mid-March to mid-May, NDVI recovered from 0.3 to 0.6. After that,
strip-grazing by cattle (Fig. 2) removed most of the recovered vegetation, as indicated by a steep decline in NDVI to about
0.3. Steady regrowth then occurred over the winter.

## 4.2 NEP and ET: fluxes, annual sums, and partitioning

Carbon flux partitioning, water input and evapotranspiration (ET) are reported as cumulative annual sums in Table 1. The
uncertainty of these annual sums is assessed separately, in Sect. 4.3.
Gross primary production of the IFR pasture (2679 g C $m^{-2}$ $yr^{-1}$) was nearly double GPP at the UUW pasture
(1372 g C $m^{-2}$ $yr^{-1}$). However, ecosystem respiration at the IFR pasture (2271 g C $m^{-2}$ $yr^{-1}$) was only 68 % greater than ER
at the UUW pasture (1352 g C $m^{-2}$ $yr^{-1}$). Consequently, NEP of the IFR pasture was 408 g C $m^{-2}$ $yr^{-1}$, as compared to
20 g C $m^{-2}$ $yr^{-1}$ of the UUW pasture (Table 1). The ratio ER/GPP was 0.85 for the IFR pasture, indicating that the ecosystem
retained a significant fraction of the C taken up from the atmosphere. This was not the case for the UUW pasture where
ER/GPP was 0.99 (Table 1).

The seasonal and diurnal variations in NEP for both pastures are illustrated in Fig. 5. For the UUW pasture, daytime net C uptake occurred only in spring and summer (November and December). For the IFR pasture, daytime uptake occurred throughout most of the year. However, longer periods of uptake are repeatedly interspersed with shorter periods of daytime C losses, visible as striations. These periods of C loss follow grazing events and correspond with periods of reduced NDVI (Fig. 4d). The repeating pattern of sign changes of NEP around grazing events is qualitatively similar to that shown by Merbold et al. (2014) around harvests. Prior to grazing, daytime NEP was greater than $1 \text{ g C m}^{-2} \text{ h}^{-1}$; after grazing, daytime NEP became negative typically for about 3 d before returning to positive values. In contrast, nocturnal NEP (C loss as respiration) appeared far less affected by grazing and ranged from 0 to $-0.6 \text{ g C m}^{-2} \text{ h}^{-1}$.

Regarding cumulative NEP (Fig. 6), the impact of each grazing at the IFR pasture appears short-term and minor compared to an overall steady rise of NEP from August, throughout most of the grazing season, until late March. Cumulative NEP of the IFR pasture levelled off at around $400 \text{ g C m}^{-2}$ between March and August and the net C loss from the final grazing event in mid-May had a more prolonged effect than all previous grazings. At the UUW pasture, NEP was initially suppressed due to a winter grazing some time before the start of our measurements. Throughout spring (Sep to Nov), NEP accumulated at rates similar to those of the IFR pasture. With the onset of summer drought in December, the UUW pasture turned into a net C source, and subsequently, NEP declined steadily, reaching a final value of $20 \text{ g C m}^{-2}$ by the end of the observation year (Fig. 6).

Cumulative ET was 785 mm for the IFR pasture, 55 % of the water input (Table 1). For the UUW pasture, annual ET was 579 mm (57 % of the rainfall). The ET patterns of the two pastures showed no significant difference until mid-December, 2.5 months into the irrigation season (Fig. 7). Until this time, cumulative ET of the UUW pasture closely matched cumulative rainfall. For the IFR pasture, water input exceeded ET from early November until early April, indicated by the increasing distance between the respective cumulative curves. From April to August, evaporation rates were similar for the two pastures and well below water input from precipitation.

Table 1 also reports the ratio of GPP to water input (WI) for the two pastures, as a measure of how efficiently the water input is used by the ecosystem to produce biomass. This ratio was $2.79 \text{ mmol C mol}^{-1} \text{ H}_2\text{O}$ for the IFR pasture and $2.03 \text{ mmol C mol}^{-1} \text{ H}_2\text{O}$ for the UUW pasture, indicating that on an annual basis, the former used its water input 37 % more efficiently than the latter.

### 4.3 Uncertainty analysis of annual budget terms

In this section, it is explained how the uncertainty estimates in Table 1 were obtained. This is first done for annual NEP. Its sources of uncertainty are listed in Table 2, with uncertainty values in absolute units. The derivation of these values is described next. The assessment of the uncertainties of GPP, ER, and the water-budget terms follows later in this section.

The first sources of NEP uncertainty are the measurement errors of the eddy-covariance method. These contain a random sampling component and a selective-sampling bias, which may arise from calibration error of the $CO_2$ analyser for certain

inter-calibration periods, or from insufficient corrections for high-frequency attenuation. Random flux measurement errors have negligible impact on the annual sums (estimated here as 0.3 % of NEP) because of the large number of contributing samples ($> 10^4$). Calibration errors were $< 1$ % as assessed by comparison of $CO_2$ concentrations to those measured with a Fourier-transform infrared spectrometer operated at the same sites (Laubach et al., 2015, this issue). High-frequency flux losses were generally small due to the high cut-off frequency of $> 1$ Hz for $CO_2$, and were corrected with the method of Fratini et al. (2012). Also, the effects of high-frequency losses for upward and downward fluxes partially cancel because both are underestimated in magnitude. Thus, we estimate the contribution of selective-sampling errors to the relative uncertainty as 2 % of annual NEP.

The next source of uncertainty is footprint error, i.e. the imperfect representation of the targeted ecosystem surface in the source area of the flux measurement. For this we identified two components: firstly, a bias from the influence of the UUW pasture on the measurements at the IFR site, and vice versa; secondly, an unknown contribution of other surfaces not representing either pasture. Details of their estimation are given in Appendix B.

The remaining sources of NEP uncertainty are those due to the gap-filling algorithm itself and to the choice of low-turbulence filter threshold, which determines the selection of input data to the gap-filling algorithm. The SOLO algorithm is bias-free in the sense that its full-year output (100 % of modelled data) has the same annual sum as the dataset merged from modelled data and existing input data (i.e. the actual output dataset of the gap-filling). However, this does not provide a measure of accuracy because both sums would be equally affected if there was model bias in the gap periods. A more realistic indication of the uncertainty of annual sums can be obtained by comparison to other gap-filling approaches (Moffat et al., 2007). To this end, we also computed NEP with the process-based gap-filling method of Barr et al. (2004). The differences in NEP between the latter method and the SOLO algorithm were obtained as 20 g C m$^{-2}$ yr$^{-1}$ for the IFR pasture and $-23$ g C m$^{-2}$ yr$^{-1}$ for the UUW pasture. We inspected visually for many larger gaps, including some grazing events, how the two gap-filling models differed. Agreement was generally excellent at night. Noticeable differences occurred mostly in daytime gaps and could be of either sign. Comparing to measured data before and after the gaps, the SOLO values generally appeared the more plausible ones. Nevertheless, we take these differences between the two gap-filling approaches as an indication of the uncertainty of the SOLO algorithm and assign it a value of 20 g C m$^{-2}$ yr$^{-1}$, for both pastures.

The uncertainty of NEP due to threshold choice is illustrated in Fig. 8, for the IFR pasture. There, NEP decreases systematically as the data-gap fraction is increased by choosing stricter (greater) threshold values for $\sigma_w$ or $u_*$. The decrease is indicated by a linear-regression line. The line is drawn only within the range of thresholds investigated here. Threshold values outside this range are not considered reasonable, and therefore, the uncertainty due to threshold choice is estimated as half the difference between the NEP values at either end of the regression line. For the IFR pasture, this yields an uncertainty of 30 g C m$^{-2}$ yr$^{-1}$ (Fig. 8), and for the UUW pasture of 8 g C m$^{-2}$ yr$^{-1}$ (not shown).

The total uncertainty of NEP results as 22 g C m$^{-2}$ yr$^{-1}$ and 38 g C m$^{-2}$ yr$^{-1}$ for the UUW and IFR pastures, respectively (Table 2). For UUW, this is dominated by the uncertainty of the gap-filling model, and for IFR by the contribution from the threshold selection. To estimate the threshold-dependent uncertainties of GPP and ER, the same procedure was followed as

for NEP. At both sites, the absolute values of GPP and ER increased systematically with increasing gap fraction (not shown). For the UUW pasture, the threshold-dependent uncertainty of GPP was 22 g C m$^{-2}$ yr$^{-1}$ and that of ER was 29 g C m$^{-2}$ yr$^{-1}$. For the IFR pasture, the respective uncertainties were 43 g C m$^{-2}$ yr$^{-1}$ for GPP and 73 g C m$^{-2}$ yr$^{-1}$ for ER. The uncertainty of the gap-filling itself is assumed, for both pastures and both variables, to be similar to that for NEP, of order

20 g C m$^{-2}$ yr$^{-1}$. The combination of the uncertainties from threshold choice and gap-filling results in relative uncertainties of order 2 to 4 % for GPP and ER. We assume another 3 % relative uncertainty for the other contributions (EC measurement and footprint). With these assumptions, we arrive at the estimates of total uncertainty given in Table 1.

To estimate the uncertainty of annual ET, the same contributions need to be considered as for NEP, but their relative weights are different. Random measurement error is again negligible. Selective-sampling bias is assumed larger for $H_2O$ than for

$CO_2$, as 5 % of annual ET. The relative contributions from footprint bias and footprint-fraction uncertainty account for less than 2 % uncertainty in annual ET. They are small for two reasons. Firstly, a noticeable difference in ET between the two pastures existed only for less than 4 months (mid-December to early April). Secondly, almost all ET occurred during daytime when the footprint was more contracted than at night. The minimal contribution of nocturnal periods to ET also implies that the gap-filling of ET is insensitive to the choice of the low-turbulence threshold. The gap-filling procedure for

ET, using NDVI and meteorological variables as drivers in a neural-network algorithm, is inherently robust and so its error contribution to annual ET less than 1 %. The uncertainty of ET is thus dominated by the selective-sampling bias of the eddy-covariance measurement. Consequently, the uncertainty values for ET in Table 1 represent 5 % of ET. Finally, the uncertainty of the water inputs is estimated from the difference between the rain gauges at the two sites, after removing the irrigation contributions for the IFR site. It is obtained as 2 % of the water input.

## 4.4 Non-$CO_2$ exchanges and NECB

The biomass DM removed by grazing from the IFR paddock was determined via empirical relationships with plate-meter-height readings. The relationships obtained individually for six sampling occasions, each associated with a grazing event, did not significantly differ from each other, which means there were no systematic seasonal differences in the way the standing

biomass compacted under the plate meter. Hence, the data were pooled and a single linear regression obtained ($R^2 = 0.85$). The uncertainty of the biomass DM estimate consists of two contributions. One is the conversion uncertainty, derived from the prediction interval of the linear regression. The other is the sampling uncertainty (stemming both from spatial variability and the random error of the plate-meter reading itself), quantified by the standard error of the 40 readings per sampling occasion. The two contributions were about equal.

The biomass removed from the IFR paddock amounted to 10,500 ($\pm$234) kg DM ha$^{-1}$ yr$^{-1}$. The weight fraction of C in the leaves and shoots of ryegrass was estimated as 41 ($\pm$1) % (Saggar and Hedley, 2001). With that, the C removed by grazing was obtained as 430 ($\pm$14) g C m$^{-2}$ yr$^{-1}$. As a consequence of grazing, the leaf area was reduced by 78 %, from 2.7 m$^2$ m$^{-2}$ to 0.6 m$^2$ m$^{-2}$.

In the UUW paddock, biomass samples were collected on 21 May 2013, 3 d prior to the winter grazing, then dried and weighed, resulting in a biomass density of 1523 ($\pm$156) kg DM ha$^{-1}$. The UUW paddock (19 ha) was then strip-grazed by 200 non-lactating cows for 9.83 d. After grazing the remaining vegetation was very short and the ground was severely pugged, so that direct biomass estimation was impossible. Instead, we employ two corroborating plausibility estimates. The first is to assume that the proportion of removed biomass is comparable to that in the IFR paddock, say 78 ($\pm$10) %. This gives 1180 ($\pm$190) kg DM ha$^{-1}$. The other is based on the feed requirement of the cows. For a non-lactating cow of 450 kg liveweight, a realistic intake is of order 10 ($\pm$2) kg DM cow$^{-1}$ d$^{-1}$ (Laubach et al., 2013; http://www.dairynz.co.nz/feed/nutrition/dry-cows/, accessed 11 January 2016). Multiplying this with the cow number and the grazing duration, 19,700 ($\pm$3900) kg DM were required. Divided by the grazed area, this yields 1040 ($\pm$208) kg DM ha$^{-1}$. Combining the two independent estimates, the biomass removed by the cows was likely to be 1,110 ($\pm$280) kg DM ha$^{-1}$. Assuming again that the average C content of the above-ground DM was 41 %, it results that grazing removed 45 ($\pm$11) g C m$^{-2}$ yr$^{-1}$. This was about 10 % of the C grazed at the IFR paddock.

The deposition of C in excreta onto the IFR pasture was calculated based on known or estimated values for $F_{grazing}$, supplemental feed, digestibility, typical urine volumes voided by cows, and the C contents of dung and urine (Appendix C). Combining C in dung and urine resulted in a total for $F_{excreta}$ of 127 ($\pm$11) g C m$^{-2}$ yr$^{-1}$ for the IFR pasture. The same estimation for the UUW pasture was simpler because the cows that grazed there during winter did not receive any supplements. The digestibility of the UUW pasture was probably lower than that of the IFR pasture due to rank growth and lack of fertiliser applications. If the digestibility was 0.70 ($\pm$0.03), then 30 ($\pm$3) % of the C ingested would have been excreted in dung. If, as for the IFR pasture, a further 3 % of the C intake was voided in urine, then the C deposited with excreta onto the UUW pasture would result as 15 ($\pm$4) g C m$^{-2}$ yr$^{-1}$.

Input of C as fertiliser ($F_{fert}$) to the IFR pasture was 7.8 g C m$^{-2}$ yr$^{-1}$, based on urea applications totalling 183 kg N ha$^{-1}$ yr$^{-1}$. The UUW pasture was not fertilised during the study year. Export of C as DOC ($F_{DOC}$) was estimated for the IFR pasture using measurements of leachate collected from similar soil, which ranged from 5 to 15 mg C L$^{-1}$ (S. M. Thomas, pers. comm., 2015). Assuming 654 ($\pm$48) L m$^{-2}$ of drainage from IFR pasture (water input minus ET, Table 1) and a DOC content of 10 ($\pm$3) mg C L$^{-1}$, then mean annual $F_{DOC}$ was 7 ($\pm$2) g C m$^{-2}$ yr$^{-1}$. For the UUW pasture, $F_{DOC}$ was likely to be even less, due to both smaller drainage and smaller C inputs than for the IFR pasture, and thus ignored. Both pastures lost a small amount of C as $CH_4$ to the atmosphere; coincidentally this amount was obtained equal for both pastures, as 3.4 ($\pm$0.3) g C m$^{-2}$ yr$^{-1}$ (Laubach et al., 2015, this issue).

Table 3 lists annual sums for the terms contributing to NECB, see Eq. (3), and their uncertainties. The NECB for the IFR pasture was 103 ($\pm$ 42) g C m$^{-2}$ yr$^{-1}$, indicating net C uptake, and was comparable to the input from excreta of 127 ($\pm$ 11) g C m$^{-2}$ yr$^{-1}$, while NEP and C removed by grazing almost balanced each other and the other contributions (leaching and fertiliser) were minor. For the UUW pasture, the removal of C by the winter-grazing slightly exceeded the C inputs from NEP and excreta deposition, resulting in a non-significant C loss, with a NECB of $-$13 ($\pm$ 23) g C m$^{-2}$ yr$^{-1}$.

## 5 Discussion

### 5.1 Carbon budget uncertainty

For both pastures, the uncertainty of annual NECB was dominated by that of NEP. The choice of filter threshold contributed significantly to the uncertainty of NEP, which is commonly the case (Papale et al., 2006). Moreover, it did so in a systematic fashion (Fig. 8). This behaviour occurred irrespective of whether $\sigma_w$ or $u_*$ was used as the threshold variable, as either choice led to the same dependence of NEP on the data-gap fraction (Fig. 8). Similar behaviour has been observed elsewhere, for forests as well as agricultural sites. For example, Anthoni et al. (2004) reported that the uncertainties in annual NEE for $\pm 0.1$ m s$^{-1}$ variation of the $u_*$ threshold ranged from 9 to 62 g C m$^{-2}$ yr$^{-1}$ across four sites in different ecosystems. The threshold-dependent variations of NEP at the IFR and UUW sites are of same sign and comparable magnitude as those of Anthoni et al. (2004). Both GPP and ER increased with increasing data-gap fraction, but the latter variable showed the stronger dependence, in line with NEP decreasing with increasing data-gap fraction. We interpret the trend in ER as the primary effect requiring a physical explanation. The trend in GPP arises by propagation, since GPP is taken as the sum of individually gap-filled time series of NEP and nocturnal ER. At our sites, nocturnal katabatic flow from the Southern Alps to the NW was common, probably displacing near-ground air with air of lower $CO_2$ concentration originating from sparser vegetation upstream. This process is likely to be the one that creates the need for the low-turbulence threshold filtering (Papale et al., 2006; Barr et al., 2013). In theory, a "threshold" that deserves its name should be well-defined in the sense that choosing a higher value would not lead to significantly different results. However, it seems that at some sites, including ours, it is difficult to specify such a threshold without ambiguity, and therefore the dependence of NEP on threshold choice contributes strongly to the uncertainty of NEP.

Another sizeable contribution to the uncertainty of NEP was the estimated gap-filling uncertainty, of 20 g C m$^{-2}$ yr$^{-1}$. This compares excellently to the value of 25 g C m$^{-2}$ yr$^{-1}$ obtained by Moffat et al. (2007) who compared various gap-filling algorithms for six European forests. The other sources of uncertainty of NEP were relatively minor for both pastures. For the IFR pasture, the total NEP uncertainty was comparable to that reported by Rutledge et al. (2015) for another dairy pasture in New Zealand, and for the UUW pasture it was comparable to that reported for a nearly C-neutral grassland (Wohlfahrt et al., 2008).

As a contributor to NECB, $F_{grazing}$ was equally important as NEP. With repeated plate-meter calibration, and sampling after grazing events every 1.5 m along four 150-m transects, we achieved a relative sampling uncertainty of order 3 %. There are no means to determine the absolute accuracy of the estimated $F_{grazing}$; instead, we show in Appendix C that, on a whole-farm basis, $F_{grazing}$ is consistent with the nutritional requirements of the cows and their milk production.

The third-largest term contributing to NECB was $F_{excreta}$. Since its estimation is directly linked to the cows' feed intake, its uncertainty is not independent of that of $F_{grazing}$. Consequently, if there was a bias in $F_{grazing}$, its effect would be proportionally reduced by a correlated bias of $F_{excreta}$. The remaining terms contributing to NECB, $F_{fert}$ and $F_{DOC}$, were independently estimated but were too small to have an impact on the uncertainty of NECB.

Our approach to treat cattle as exporters of biomass-C and importers of excreta-C differs from other studies of grazed grasslands, where the cattle are considered as part of the ecosystem (Soussana et al., 2007a; Rutledge et al., 2015). The management practices of this dairy ecosystem were well-suited to this approach, and uncertainty around the spatial distribution of cattle respiration during grazing was avoided. Since C respired by cattle originated from C in the $F_{grazing}$ term in Eq. (3), it had to be excluded from the C budgets to avoid double-counting. This makes it difficult to compare our NEP results directly with the majority of grazed-grassland sites, where NEP includes cattle respiration (Soussana et al., 2007a; Rutledge et al. 2015). However, the NECB results can still be compared directly.

## 5.2 Effects of management practices

The two pastures used in this study were managed differently with respect to irrigation, fertiliser application and grazing (Fig. 2), and so while differences in NEP and NECB may be in response to management overall, it is not possible to tease apart the effect of its individual components. Until mid-December, management made no difference on cumulative ET (Fig. 7), and little on cumulative NEP. By this point in time, the UUW pasture production (initially suffering from the impact of a winter-grazing that had occurred before measurements started) had caught up with the IFR pasture, both reaching about 290 g C m$^{-2}$ (Fig. 6). From mid-December onwards, the UUW pasture turned from a C sink into a C source, as a consequence of the depleted soil water content (Fig. 4b), while the intensively-managed IFR pasture remained fully productive throughout summer and evaporated at higher rates than the UUW pasture. Clearly, the main effect of IFR-pasture management was to counteract water stress as a principle controlling factor in NEP. Further, the additional water input from irrigation was efficiently used, since the ratio of GPP to total water input for the IFR pasture (2.79 mmol C mol$^{-1}$ H$_2$O) was significantly greater than for the UUW pasture (2.03 mmol C mol$^{-1}$ H$_2$O), by 37 % (Table 1).

It is instructive to compare the partitioning of NEP. For the IFR pasture, GPP was 95 % greater than for the UUW pasture, while ER was only 68 % greater (Table 1). Thus, additional C uptake outpacing additional respiration explains the stronger C sink at the IFR pasture. From a management perspective, the additional water and fertiliser inputs were highly efficient, by allowing to nearly-double the production of biomass, but only a third of the additional C removed from the atmosphere was converted into feed intake for the cows and two-thirds were respired. In part, the more muted response of ER to management (compared to GPP) may be linked to lower soil temperatures during summer (Fig. 4a), as an indirect effect of irrigation (Rajan et al., 2013). To fully understand the fate of C in managed pasture systems, the processes linking the distribution of soil water and nutrients with respiration need to be studied (Conant et al., 2001; Soussana et al., 2007b; Rutledge et al., 2015).

Carbon removed as a result of grazing at the IFR pasture was approximately equal to NEP, so the system was almost carbon-neutral with respect to production and biomass-export. Altering the number of grazing events and total biomass C exported would have changed the annual NECB, but it is not known to what extent grazing directly affected GPP. The pattern of brief, high-density grazing events and subsequent re-growth, evident in the saw-tooth pattern in NEP (Fig. 6), indicated that the

reduction of leaf area was followed by a drop and then a rise in production. This is typical of grassland sites with repeated grazing or harvest events (Wohlfahrt et al., 2008; Merbold et al., 2014). During the growing season, repeated defoliations kept the grass in a permanent juvenile state, which helped to maintain strong biomass growth throughout. Within one week after a grazing event, daily average NEP values changed from negative to positive. This resilience to grazing is one factor contributing to the positive NECB of the IFR pasture. The other crucial factor is that cattle excreta were returned to the pasture. Without excreta inputs, NECB would have been near-zero or slightly negative (Table 3). Cooper et al. (2011) found that manure additions for 17 years enhanced SOC stocks without enhancing C mineralisation rates. If this were the case for the IFR pasture (and if mineralisation rates did not increase) then SOC would be expected to accumulate over time. Additional investigations into the fate of cattle excreta for building soil C under intensively managed pastures are warranted.

## 5.3 Comparison to other studies of NECB of grasslands

The results from our study compare well to another paired-site eddy-covariance study of managed grasslands, in Switzerland (Ammann et al., 2007). Annual precipitation and mean temperature at our site and that Swiss site are quite similar; however, our site differs in that it typically has very little or no snow cover, unlike the Swiss site. There, one pasture was repeatedly fertilised and the other was not, and the former was harvested more frequently than the latter. No grazing or irrigation occurred at these sites. For the intensively-managed pasture, the mean NECB over 3 years was 147 ($\pm$130) g C m$^{-2}$ yr$^{-1}$, very similar to the NECB of the IFR pasture. For the extensively-managed pasture, Ammann et al. (2007) reported a non-significant net C loss, similar to the UUW pasture. Our NECB results are also compatible with those of a replicated small-plot study of four types of grassland management in Wisconsin, where NECB for the intensive rotational-grazing treatment (without irrigation) varied between years within $\pm$100 g C m$^{-2}$ yr$^{-1}$ and was consistently larger than for three other treatments, including unmanaged pasture (Oates and Jackson, 2014).

Rutledge et al. (2015) collected available annual C budgets of temperate grazed grasslands from the literature and displayed the relationship of NECB versus NEP for these. For the majority of site-years, both variables were positive. The IFR pasture would fit well into the main trend of the figure of Rutledge et al. (2015), after redefining NEP for this purpose so that it included cattle respiration. Such a redefined NEP would be reduced compared with our NEP excluding cow respiration (408 g C m$^{-2}$ yr$^{-1}$). The reduction would need to be of order 50 % of the C intake from grazing (thus, ca. 215 g C m$^{-2}$ yr$^{-1}$), by inference from the C fractions converted to milk and excreta adding to about 45 % of C intake (Appendix C) and C emitted as CH$_4$ accounting for about 4 % (MfE, 2014). Hence, the redefined NEP would amount to slightly less than 200 g C m$^{-2}$ yr$^{-1}$. When compared to the other managed temperate grasslands, both NEP and NECB for the IFR pasture were quite substantial, which shows that good management can result in both large pasture production and soil C storage. By contrast, the results for the UUW pasture would fall close to the origin of the figure of Rutledge et al. (2015). This is at the margin of the data distribution, reflecting the fact that this pasture was only weakly managed.

The NECB result for the IFR pasture does not agree with soil-core studies that estimate changes in C stocks at decadal time-scales (Kelliher et al., 2012; Schipper et al., 2014). Carbon stock studies based on soil core data indicated substantial C losses for some intensively-managed dairy pastures in New Zealand (Schipper et al., 2014). Decadal-scale SOC stock studies often lack detailed information about management activities (sowing, irrigation, fertiliser application and grazing) that are very relevant to the net C source or sink strength. These SOC stock data provide a longer view, while the method described here provides farm managers with an opportunity to review decisions regarding the ecosystem C balance. Here, we point to the importance of understanding seasonal water and carbon dynamics and potential mechanisms for building SOC through management. However, it is uncertain if the IFR pasture will remain a C sink in the long-term (Smith, 2014). In particular, the common practice of pasture renewal every few years may impact on the long-term NECB of managed pastures. Coupling eddy-covariance and SOC stock studies (Leifeld et al., 2011) may be the way forward to understand both short- and long-term effects of pasture management on SOC.

## 6 Conclusions

The intensively-managed (IFR) dairy pasture retained significantly more C than it lost, despite multiple grazing events, while the only weakly managed UUW pasture was C-neutral. These results are in good agreement with those of another paired grassland study on management intensity, by Ammann et al. (2007). Of crucial importance for the positive NECB of the IFR pasture was the C input from excreta. Without this (as in a cut-and-carry management system), the IFR pasture would have been C-neutral or subject to small C losses. It is unknown whether the IFR pasture will remain a C sink in the long-term, but over the study year it functioned as a highly productive, carbon-accumulating unit.

The combined management effects of the IFR pasture increased both the biomass production (GPP) and respiration (ER) significantly, compared with the UUW pasture, and the increase in GPP outpaced the increase in ER. At the same time, the ratio of GPP to water input for the IFR pasture was significantly greater than for the UUW pasture, indicating that the addition of irrigation water also boosted the production benefit of the natural precipitation. We may caution that such efficient water usage requires careful application management, based on the monitoring of soil water content.

Scientifically, this study indicates a need to further investigate the soil processes that control ecosystem respiration under irrigation, in order to fully understand whether, and how, the long-term building of carbon stocks under pasture can be achieved.

**Data availability**

The data are available on the OzFlux Portal http://data.ozflux.org.au/portal/pub/listPubCollections.jspx, under the Collection name "Beacon Farm". The citation of the data collection is:

Laubach, J. (2016): Beacon Farm OzFlux: Australian and New Zealand Flux Research and Monitoring. hdl: 102.100.100/26730.

**Appendices**

**Appendix A. Energy budget closure**

The surface energy budget was examined as a means of identifying potential site- or instrumentation-specific biases of the eddy-covariance (EC) measurements (Aubinet et al., 2000). The degree of closure (DC) is commonly reported as the slope of a linear regression (Wilson et al., 2002), with the sum of sensible and latent heat fluxes as the dependent variable and the "available energy" as the independent variable. Available energy is the incoming net radiation minus ground heat flux and heat storage in air and vegetation below the measurement height. Based on half-hourly values, the annual DC was 0.89 and 0.91 at the UUW and IFR sites, respectively (with very small regression intercepts, of $+2$ and $-4$ W m$^{-2}$, respectively). If the energy stored chemically as the result of photosynthetic activity is included in the calculation, both DC values increase by only 0.01. The DC values for both sites are closer to the ideal value of 1 than the average DCs found in all published FLUXNET intercomparison studies to date, see Stoy et al. (2013) and references therein. Compared to other low-vegetation experiments, the DC values are close to the median (0.9) found by Laubach and Teichmann (1999) reviewing 11 studies over grass or crops, close to the grassland average (0.86) reported by Stoy et al. (2013), and comparable to those for three Australian grasslands (0.87, 0.86 and 0.94) given by Leuning et al. (2012). To conclude, the surface energy budgets at our sites give no indication of any quality issues beyond what is commonly accepted as inevitable methodical error of the EC technique.

**Appendix B. Footprint bias of $CO_2$ fluxes and its correction**

During periods of northerly winds, the $CO_2$ exchange of the UUW pasture could contribute to, or "contaminate", the $CO_2$ flux measured at the IFR site, and vice versa for southerly winds (Fig. 1). The small ratio of height to fetch (distance to fence) in our setup ensured that the target pasture was always the dominant contributor to the flux. To assess the minor contributions of areas farther upwind, we used the footprint tool of Neftel et al. (2008), which is a spatially explicit implementation of the model of Kormann and Meixner (2001). In addition to the two pastures under investigation, the circular irrigated area of the neighbouring farm to the west was specified as a third source area, with the assumption that its

$CO_2$ exchange equalled that from the IFR pasture. The sum of the footprint fractions from the three specified source areas was generally < 1, with a median of 0.97. The "missing" 3 % of source area contributions can be attributed to surfaces farther afield, which are slightly overestimated by the model (Neftel et al., 2008). To assess the footprint error from the specified source areas only, the "known bias", the footprint fractions from the three specified source areas were re-normalised so that their sum equalled 1 in each run. The "missing" 3 % was taken as the best available estimate of the "unknown bias" (from surrounding areas of unknown $CO_2$ exchange rates).

For evaluating the flux uncertainty due to footprint effects, runs with gap-filled flux values are irrelevant. Most occurrences of strongly stable stratification, when footprints are largest, were excluded as a consequence of the low-turbulence threshold filtering.

At the UUW site, the sum of the re-normalised footprint fractions from the two irrigated circles (IFR and neighbouring farm) was often negligible, because westerly and southerly winds were infrequent. During the day, the irrigated pastures contributed < 0.01 in 90 % of runs and always < 0.08, and at night < 0.01 in 85 % of runs and always < 0.23. At the IFR site, the re-normalised footprint fraction from the UUW pasture during the day was < 0.01 in 82 % of valid daytime runs and always < 0.10. At night, it was < 0.01 in 65 % of runs, < 0.07 in 95 % of runs, and always < 0.25.

The footprint model results were analysed statistically, for each of the four seasons, and separately for day and night (loosely associated with unstable and stable stratification of the surface layer, which are parameterised differently in the footprint model). For each of these eight classes, we binned both the footprint fractions and the resulting net flux corrections by wind direction. For the UUW site, the bin-means of the footprint fractions of the irrigated pastures were < 0.02 most of the time. The only exceptions occurred during winter, for wind directions from 200° to 280°, with footprint fractions ranging from 0.04 to 0.12 at night and from 0.02 to 0.07 during the day. This seasonal pattern can be attributed to a higher prevalence of stable stratification in winter, for both night- and daytime. For the IFR site, the bin-means of the footprint fractions of the UUW pasture were very consistent throughout spring, summer and autumn, exceeding 0.01 only for the wind direction sector from 290° to 30°, with maximum bin-means of 0.04 during the day and 0.06 at night for northwest winds. In winter, the directional sector of UUW influence extended farther to southwest (240° to 30°) and maximum bin-means were twice as large as in the other seasons. Again, the seasonal pattern can be explained by more common occurrences of stable stratification in winter.

The effect of the footprint fractions on the $CO_2$ fluxes was assessed as follows. For each run with valid fluxes, the fluxes at the two sites were linearly combined, weighted by their respective footprint fractions, to give $CO_2$ emission/uptake rates for the two underlying pastures (UUW and IFR). This procedure was equivalent to the matrix approach of Mukherjee et al. (2015). At each site, the uncorrected nighttime and daytime fluxes were summed separately, and so were their footprint-corrected counterparts. The differences between corrected and uncorrected sums gave the nighttime and daytime footprint biases. At the UUW site, both nocturnal $CO_2$ emission and daytime $CO_2$ net uptake in winter were overestimated without footprint correction. Since nocturnal $CO_2$ emissions were overestimated by a larger fraction than daytime $CO_2$ net uptake, the annual NEP of the UUW pasture was slightly underestimated, by 0.7 g C m$^{-2}$ yr$^{-1}$. At the IFR site, the biases resulting from

the UUW footprint contributions were 12.1 and −4.8 g C m$^{-2}$ yr$^{-1}$ for nocturnal $CO_2$ emission and daytime $CO_2$ net uptake, respectively. This means that without footprint correction, both processes were underestimated, but the nocturnal respiration bias was larger, and hence NEP was overestimated by 7.3 g C m$^{-2}$ yr$^{-1}$. This may appear surprising, but at night the source area extends farther than during the day, and the winter season contributed most to the footprint bias, when the photosynthetic uptake rates were smallest.

## Appendix C. Carbon inputs and outputs of the dairy herd

In the main text, we report the measurements of the biomass removed by grazing, per area, and its C content. This appendix serves the following purposes: the per-area estimates are related to estimates per cow and per day, it is described how C deposition with excreta is linked to the biomass intake (from pasture and supplements combined), the C content of the milk produced is calculated, and it is assessed whether all these estimates are consistent and realistic. A full C budget of the dairy herd is not constructed, as that would require independent estimates of the cows' $CO_2$ respiration, $CH_4$ emissions, and liveweight gains, which were not obtained here.

### C1. Seasonal budget and area considerations

The milking season was 248 d long (25 Sept 2012 to 29 May 2013). There were 868 cows, managed in two herds. The total irrigated-pasture area of the farm was 328 ha. Dividing this area by the previous two numbers gives a factor, $S$, which converts from amounts per grazing area per year to amounts per cow per day. This factor was thus $S = 15.25$ m$^2$ yr cow$^{-1}$ d$^{-1}$.

### C2. Biomass removed by grazing

The biomass dry-matter (DM) removed by grazing from the IFR pasture was determined as 10,500 ($\pm$234) kg DM ha$^{-1}$ yr$^{-1}$, and the C contained therein as 430 ($\pm$14) g C m$^{-2}$ yr$^{-1}$ (Sect. 4.4 of main text). Using the conversion factor $S$, each cow consumed 16.0 ($\pm$0.35) kg DM cow$^{-1}$ d$^{-1}$ from the IFR pasture, containing 6.56 ($\pm$0.21) kg C cow$^{-1}$ d$^{-1}$.

### C3. Supplementary feed and total feed intake

The cows were fed 1.8 ($\pm$0.2) kg DM cow$^{-1}$ d$^{-1}$ of barley supplements at the milking shed (11 % of the grazed DM). Assuming 40 % C content, the cows would have received 0.72 kg C cow$^{-1}$ d$^{-1}$ from supplements, equivalent to 47 g C m$^{-2}$ yr$^{-1}$. Grazed biomass and supplements combined to 17.8 ($\pm$0.4) kg DM cow$^{-1}$ d$^{-1}$, which is a realistic value, in line with nutritional recommendations for highly-productive Holstein-Friesian/Jersey crossbreds of 450 kg liveweight

([http://www.dairynz.co.nz/feed/nutrition/lactating-cows/](http://www.dairynz.co.nz/feed/nutrition/lactating-cows/), accessed 11 January 2016). Expressed per grazing area, the total DM intake (DMI) was 11,670 ($\pm$260) kg DM ha$^{-1}$ yr$^{-1}$.

## C4. Carbon content of excreta

A fraction $1 - D$ of the DMI is excreted as dung, where $D$ is digestibility. Monthly averages for $D$ from 10 dairy farms across New Zealand for the milking season (September to May) range from 0.74 to 0.81, with a mean of 0.77 (MfE, 2014, Table A3.1.6). Using the mean value here, 23 % of total DMI was excreted, equal to 2685 kg DM ha$^{-1}$ yr$^{-1}$. Assuming a C content of dung of 41.5 % (van der Weerden et al., 2014), the cows excreted 111 g C m$^{-2}$ yr$^{-1}$, equivalent to 1.70 kg C cow$^{-1}$ d$^{-1}$, while grazing IFR pasture (which amounts to 25.9 % of the C intake from grazing alone). The uncertainty of the excreted C combines the relative errors of DMI (2.3 %) and of $1 - D$ (8 %) and results in 9 g C m$^{-2}$ yr$^{-1}$. The C content of cow urine can vary considerably. Based on the measured values of 7.5 ($\pm$1.2) g C L$^{-1}$ (Lambie, 2012) and 15.9 ($\pm$0.75) g C L$^{-1}$ (Lambie et al., 2012) we estimate it here as 12 ($\pm$4) g C L$^{-1}$. With typically 10 ($\pm$1) urinations per cow per day (Haynes and Williams, 1993; White et al., 2001), and assuming a typical volume of 2 L urine per urination, the excreted urine-C was 0.24 ($\pm$0.08) kg C cow$^{-1}$ d$^{-1}$, about 14 % of the dung-C and 3 % of the cows' C intake. Dividing by $S$, this converts to a deposition rate of 16 ($\pm$5) g C m$^{-2}$ yr$^{-1}$. The amounts of C in dung and urine deposited on the IFR pasture add to a total of 127 ($\pm$11) g C m$^{-2}$ yr$^{-1}$.

## C5. Carbon in milk

Based on management records, the amount of milk produced at the farm during 2012/13 was 3.96 * 10$^6$ L, or 4.08 * 10$^6$ kg based on a density of 1.03 kg L$^{-1}$. On a per-animal, per-day basis (divided by 868 cows and by 248 d) the average milk production was  18.9 kg milk cow$^{-1}$ d$^{-1}$. According to Wells (2001), cow milk in New Zealand should contain on average 0.0684 kg C kg$^{-1}$ milk. By contrast, recent repeated measurements using milk from a Jersey-cow herd in the North of New Zealand yielded a mean of 0.085 kg C kg$^{-1}$ milk (S. Rutledge, pers. comm., 2016). Here we take both figures as estimates of the range of this variable; accordingly, the average milk-C production was 1.29 to 1.61 kg C cow$^{-1}$ d$^{-1}$. This represents 18 to 22 % of the total C intake (from grazing and supplements), which is in excellent agreement with Soussana et al. (2010), who assumed for intensive dairying that 19 to 20 % of the C intake ended up in the milk produced. This agreement suggests that the milk production data and our estimate of grazed biomass-C are consistent, and we view this consistency as further confirmation of the validity of the biomass data.

## Acknowledgements

The authors thank Synlait Farms Co. (now Purata) for supporting our research, in particular Lucy Johnson for valuable help with locating a suitable site and ongoing communication. The farm managers, Leith Norton and Glenn McCallum, provided valuable information. Thanks are due to David Whitehead for initiating the project, to Graeme Rogers and Tony McSeveny for technical support in the field, and to Gerardo Fratini (LI-COR Biosciences Inc.) for providing a corrected version of the EddyPro software. We are indebted to Peter Isaac, Paul Mudge and Susanna Rutledge for many useful discussions. This research was undertaken with CRI Core Funding from New Zealand's Ministry of Business, Innovation and Employment.

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

**Tables**

Table 1. Annual $CO_2$ budget and water budget terms and their ratios for the UUW pasture (unirrigated, unfertilised, winter-grazed) and the IFR pasture (irrigated, fertilised, rotationally-grazed). Estimated uncertainties are given in parentheses.

| | | UUW | IFR |
|---|---|---|---|
| $CO_2$ budget terms | NEP | 20 (22) | 408 (38) |
| (g C m$^{-2}$ yr$^{-1}$) | GPP | 1372 (51) | 2679 (108) |
| | ER | 1352 (54) | 2271 (102) |
| $H_2O$ budget terms | WI | 1014 (20) | 1439 (28) |
| (mm yr$^{-1}$) | ET | 579 (29) | 785 (39) |
| Ratios | ER/GPP | 0.99 (0.05) | 0.85 (0.05) |
| | ET/WI | 0.57 (0.03) | 0.55 (0.03) |
| | GPP/WI (mmol C mol$^{-1}$ $H_2O$) | 2.03 (0.09) | 2.79 (0.12) |

NEP, net ecosystem productivity; GPP, gross primary productivity; ER, ecosystem respiration; WI, water input (precipitation + irrigation); ET, evapotranspiration.

**Table 2. Uncertainty estimates (g C m$^{-2}$ yr$^{-1}$) of net ecosystem production (NEP) for the UUW and IFR pasture. The total uncertainty is calculated as the root of the sum of the squares of the listed contributions (excluding footprint bias).**

| Uncertainty source | UUW | IFR |
| --- | --- | --- |
| Random sampling error | < 0.1 | 1.2 |
| Selective-sampling bias | 0.4 | 8 |
| Footprint bias | −0.7 | +7.3 |
| Footprint fraction uncertainty | 3 | 10 |
| Gap-filling model uncertainty | 20 | 20 |
| Nocturnal threshold choice | 8 | 30 |
| Total uncertainty | 22 | 38 |

**Table 3. Carbon gains and losses (g C m$^{-2}$ yr$^{-1}$), and their uncertainties in parentheses, for the UUW and IFR pasture, including NEP, grazing ($F_{grazing}$), animal excreta ($F_{excreta}$), fertiliser ($F_{fert}$), leaching of dissolved organic C ($F_{DOC}$), and methane from soil and excreta ($F_{CH4}$). Negative numbers indicate a loss of carbon from the pasture ecosystem.**

| C budget term | UUW | IFR |
|---|---|---|
| NEP | 20 (22) | 408 (38) |
| $+F_{excreta}$ | 15 (4) | 127 (11) |
| $+F_{fert}$ | 0 (0) | 8 (1) |
| $-F_{grazing}$ | −45 (11) | −430 (14) |
| $-F_{DOC}$ | nd | −7 (2) |
| $-F_{CH4}$ *) | −3.4 (0.3) | −3.4 (0.3) |
| NECB | −13 (23) | 103 (42) |

NEP, net ecosystem productivity; NECB, net ecosystem carbon budget; nd, not determined.

*) The values for $F_{CH4}$ originate from Laubach et al. (2016). They are given in their Table 3, in different units (8.9 (±0.79) nmol CH$_4$ m$^{-2}$ s$^{-1}$). That table also shows that the underlying data series for the two sites differ from each other,
10 and that the numerical equality of the annual estimates using the "merged daily" approach occurred by happenstance.

**Figures**

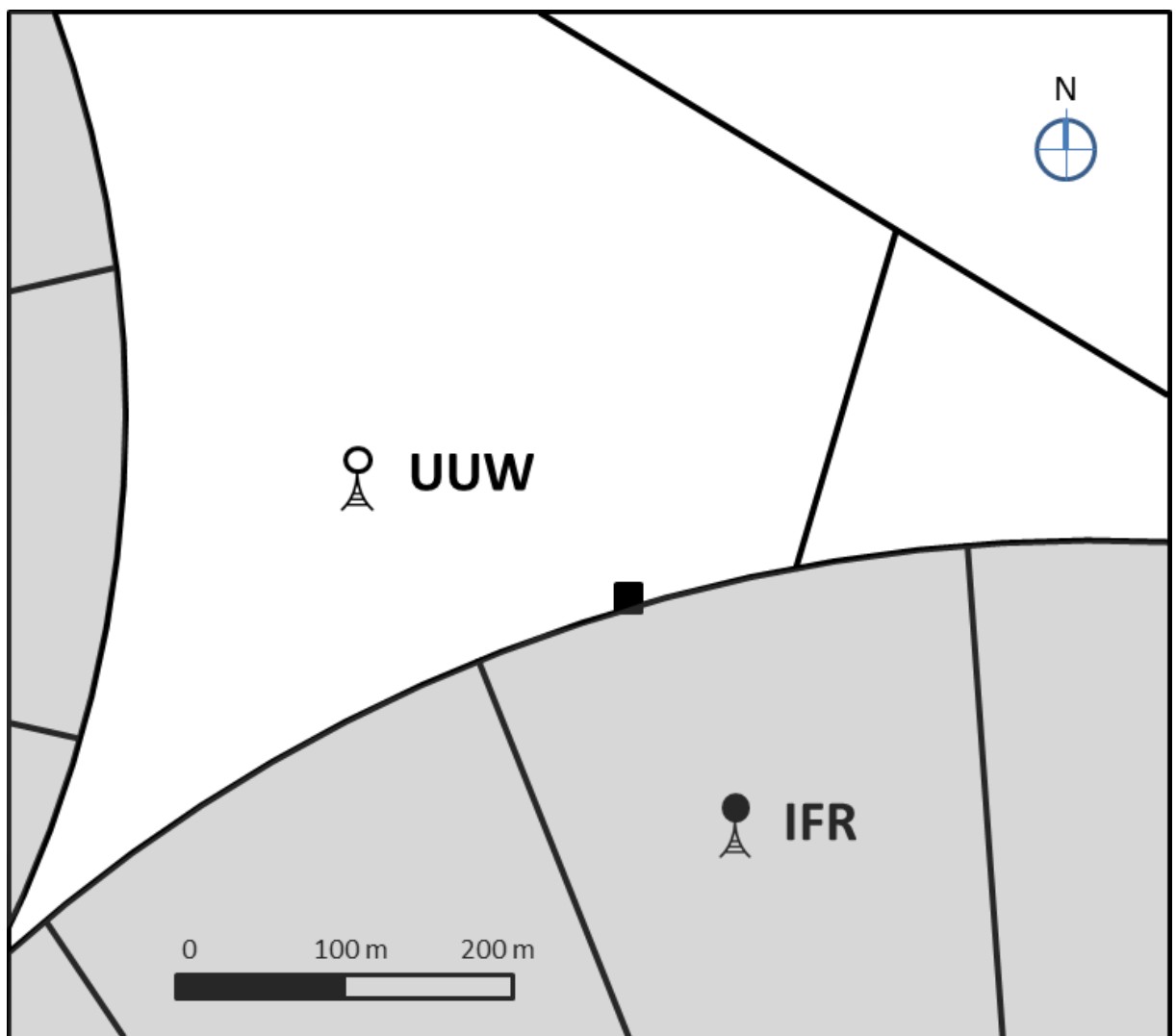

Figure 1. Schematic of paddocks and instrument layout on a commercial dairy farm on the Canterbury Plains of New Zealand. Areas of irrigated, fertilised and rotationally grazed pasture (IFR) are shaded in grey. An area of unirrigated, unfertilised and winter-grazed pasture (UUW) is wedged between two pivot-irrigation circles. The measurement sites in the IFR and UUW paddocks are indicated by mast symbols. An instrument hut was situated on the boundary between the two paddocks (filled square).

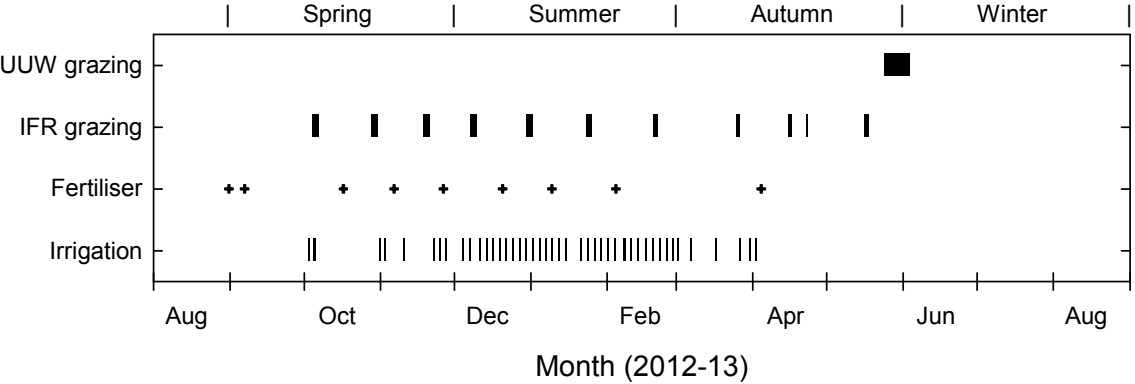

**Figure 2. Pasture management regime from 17 August 2012 to 16 August 2013. Bars indicate the time and duration of grazing events in the UUW and IFR paddock. Crosses and vertical lines mark times of fertiliser and irrigation applications, respectively, to the IFR paddock.**

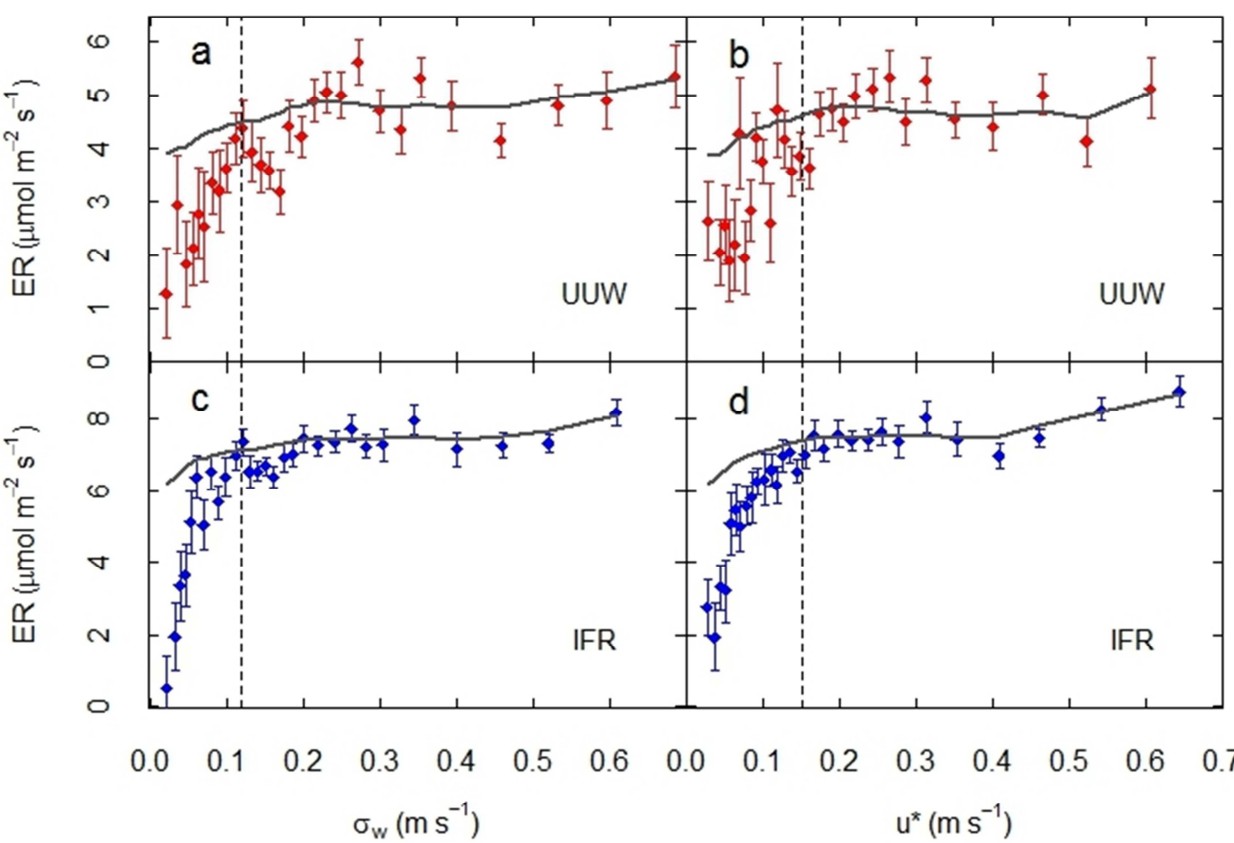

**Figure 3.** Example of moving-point threshold detection for night-time $CO_2$ fluxes (*ER*), for the temperature class 13 to 16 °C (soil temperature at 20 mm depth), at the UUW site (top) and the IFR site (bottom). In Panels a and c the standard deviation of vertical wind speed ($\sigma_w$) is used as the discriminating variable, following Acevedo et al. (2009); in Panels b and d, friction velocity ($u^*$) is used instead, as is common practice. Bin means and their standard errors are indicated by dots with error bars. The solid line marks 0.99 times the mean of all bin-means to the right of the actual bin. The vertical dashed lines indicate the overall optimum threshold choices for $\sigma_w$ and $u^*$, respectively.

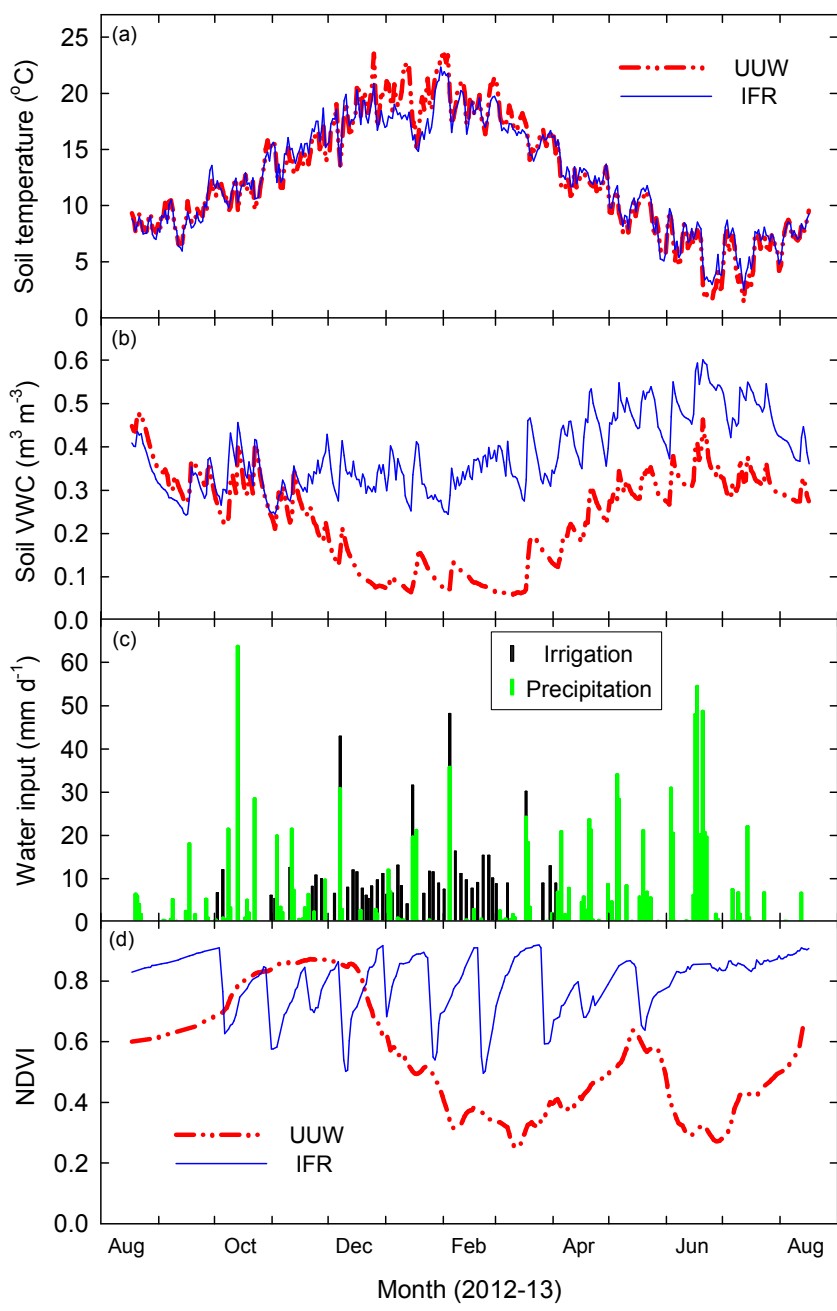

**Figure 4. Time series at the UUW and IFR sites for daily means of (a) soil temperature and (b) soil moisture at 50 mm depth; (c) daily sums of rainfall (both sites) and irrigation (IFR only); and (d) daily values of the Normalised Difference Vegetation Index (NDVI).**

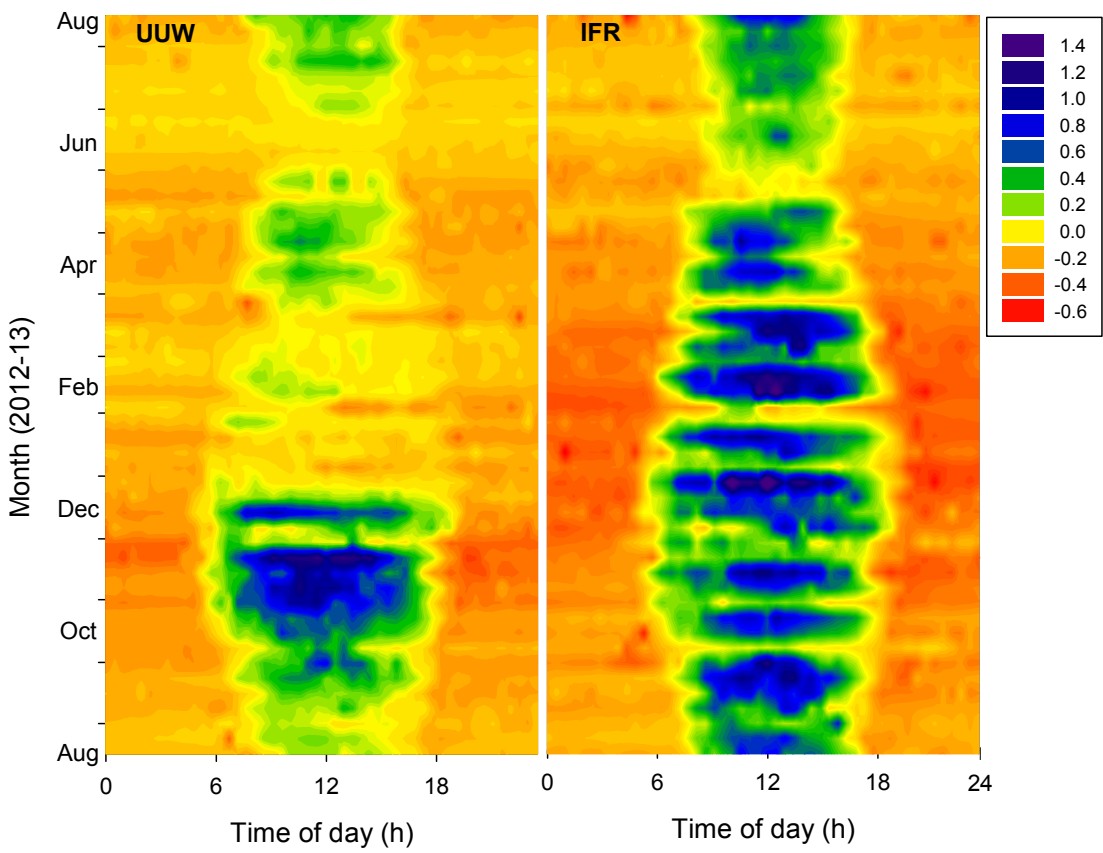

**Figure 5. Visualisation of diurnal variation (horizontally) and seasonal variation (vertically) of net ecosystem production (g C m$^{-2}$ h$^{-1}$) for the UUW and IFR pastures. Positive values (in yellow, green and blue) represent net C uptake by the ecosystem, and negative values (in orange and red) represent net C losses.**

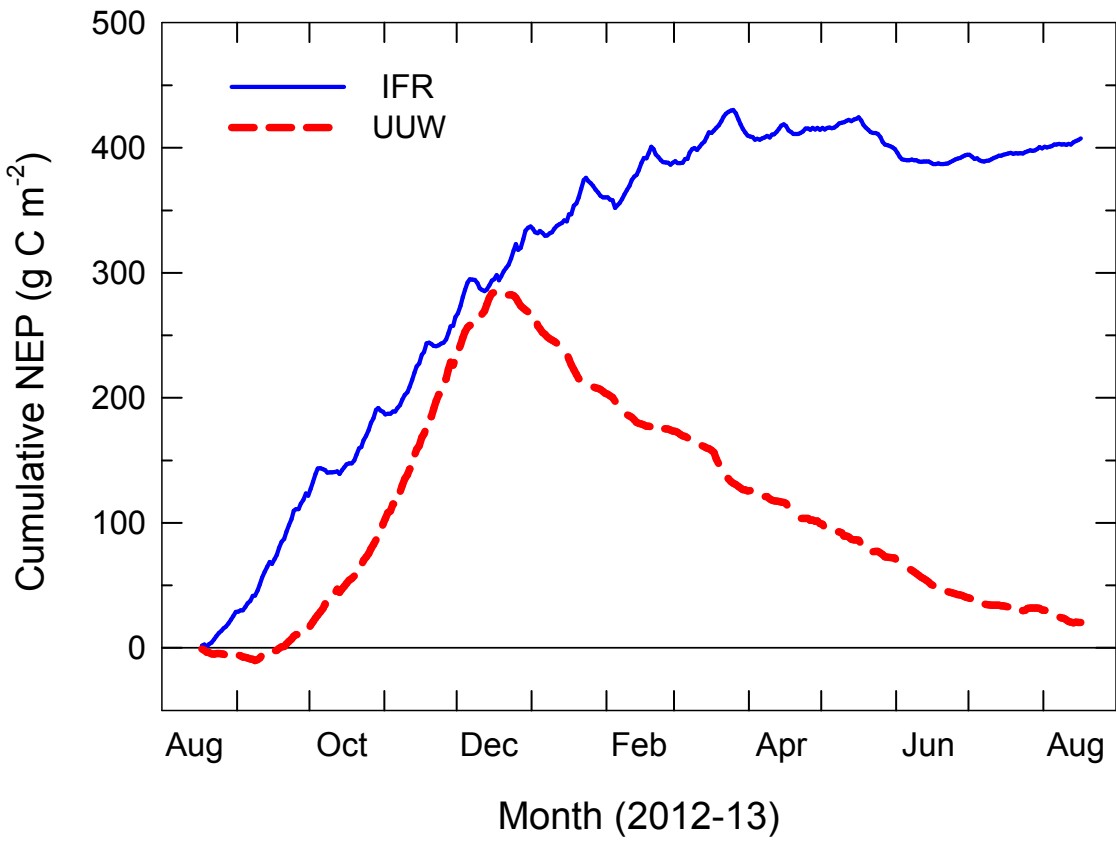

**Figure 6. Cumulative net ecosystem production for the irrigated, fertilised and rotationally grazed (IFR) and the unirrigated, unfertilised and winter-grazed (UUW) pasture. Positive values represent C uptake by the ecosystem.**

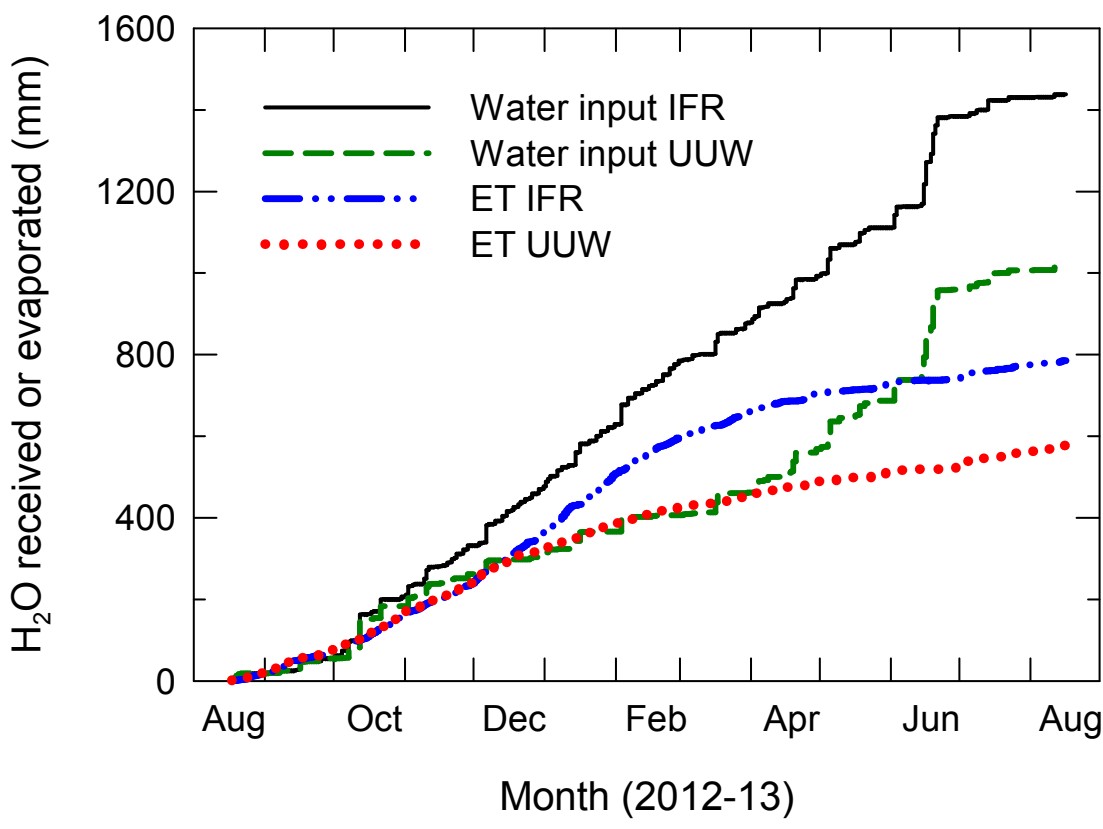

**Figure 7. Cumulative water input (sum of rainfall and irrigation) and evapotranspiration (ET) for the IFR and UUW pasture.**

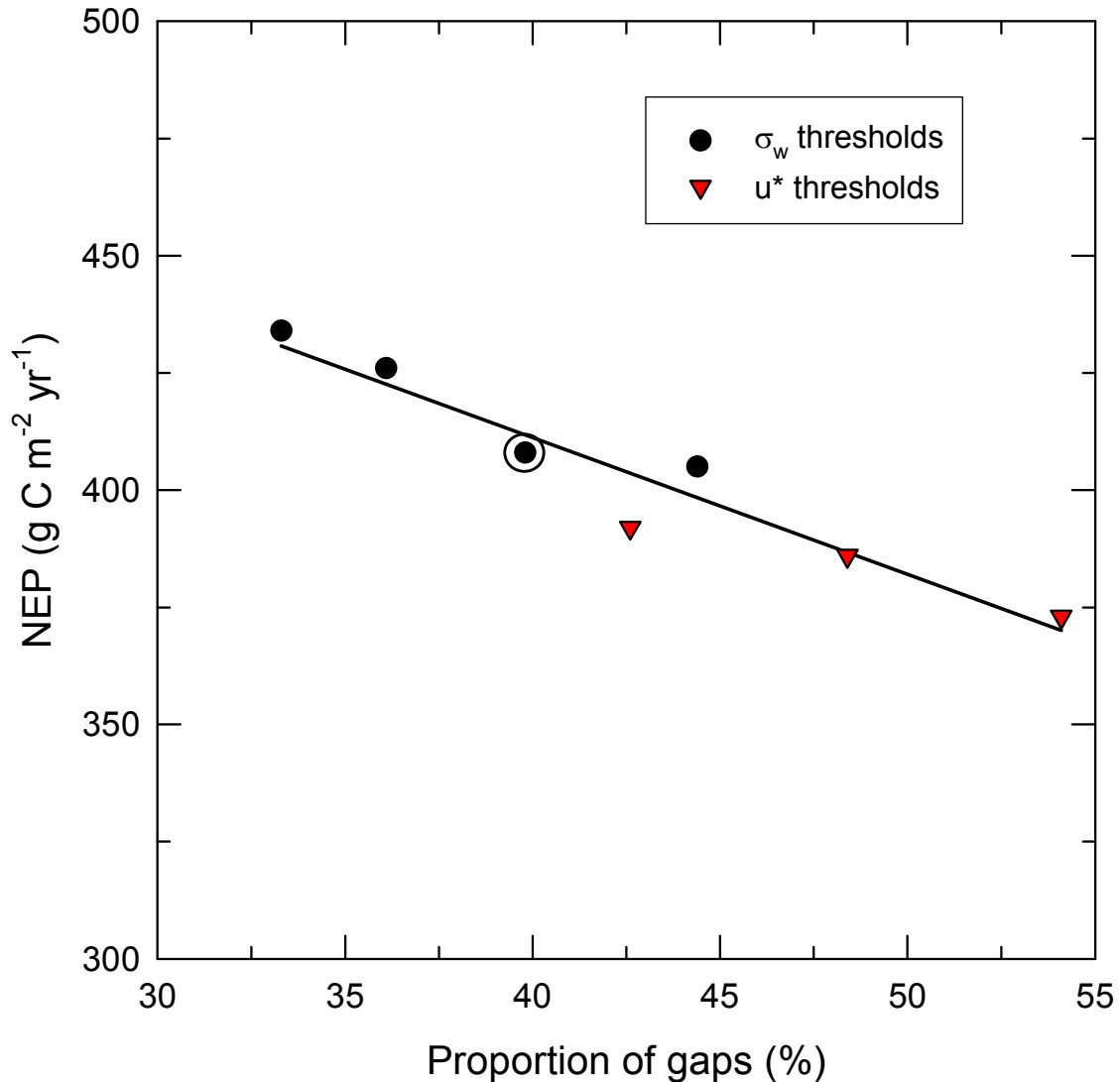

**Figure 8. Variation of annual NEP estimate for the IFR pasture with the fraction of data gaps, as resulting from the choice of low-turbulence filter threshold. The $\sigma_w$ thresholds (dots) are, from left to right, 0.08, 0.10, 0.12 and 0.15 m s$^{-1}$; the $u_*$ thresholds (triangles) are 0.12, 0.15 and 0.18 m s$^{-1}$. The optimal threshold, based on moving-point threshold detection, is indicated by a circle. The solid line, obtained by linear regression, is displayed only to indicate the qualitative trend.**