# Peer review of "Carbon budgets for an irrigated intensively-grazed dairy pasture and an unirrigated winter-grazed pasture"

_Biogeosciences, 2016_

## Referee Comment (RC1) · Anonymous Referee #1 · 11 Mar 2016

Review of bg-2016-46: "Carbon budgets for an irrigated intensively-grazed dairy pasture and an unirrigated winter-grazed pasture."

In this manuscript, the Authors investigate the carbon (C) exchange dynamics and C balance of cattle-grazed pastures in New Zealand using the eddy covariance (EC) technique, and a technique to estimate C uptake and loss by cattle. As is the case with many grasslands and pastures globally, a mostly-ungrazed pasture was net C neutral during the one-year study period, whereas a pasture that experienced irrigation, fertilization and periodic grazing was a net C sink. Yet, the C sink dynamics of this intensively-managed pasture hinged on whether or not C uptake and loss by cattle was accounted for in the ecosystem C balance.

This manuscript is well-written, the C measurement methodology is technically-sound, and I have only nominal contextual comments. However, I am concerned that this study does not present an advancement in understanding of pasture C dynamics, nor is it up to par with other, similar studies. Specifically, I am concerned that the Authors attempt to draw conclusions on ecosystem C balance using a single year of measurements, especially because there is no true control ecosystem nor is there any pretreatment comparison. Yet, for this OZFlux special issue, I think this manuscript may ultimately be considered for publication because it focuses specifically on management issues in New Zealand. I will be happy to recommend this article for publication once my main concerns are addressed.

1) The Authors need to better frame their findings in the context of existing literature on pasture/rangeland C dynamics and C balance, and also the C balance of grasslands in general. I believe that this study is best presented as a supplement to the larger body of literature that exists on this subject. It may also be useful to focus more on the weekly and seasonal dynamics of C exchanges as a way to differentiate this manuscript from longer-term studies.

Potential sources include:

Felber et al. 2016. Agricultural and Forest Meteorology
McGinn et al. 2014. Journal of Environmental Quality
Oates and Jackson. 2014. Rangeland Ecology and Management

2) Because the UUW pasture is not a true control, the Authors may wish to truncate their study period to Aug 2012 – May 2013 (so grazing did not occur in the UUW pasture during the study period). Alternatively, it may be appropriate to cite literature on the neutral C balance of other ungrazed grasslands in NZ, which would support the Authors' determination that the grassland in their study is an acceptable control.

3) On page 16, line 15, the Authors state that the UUW pasture and IFR pasture had different grazing histories prior to the study period. Without a pretreatment comparison, I am greatly concerned that these pastures aren't comparable.

In-text comments:

I recommend that the Authors use active voice throughout the manuscript.

I recommend using negative NEE values to indicate a carbon sink or uptake by the ecosystem, and positive NEE values to indicate a carbon source or loss.

Line 11: Remove sentence beginning, "Primary terms..."

Line 17: Differences in GPP and RE are both very large. I recommend simply stating these differences as a result.

Line 18: Efficiency measured as what metric or variable?

Line 19: Need a stronger conclusion than this. What new information was obtained in this study?

Line 24: Intensification of grasslands needs changed.

Line 26: Needs citation.

Line 29: Remove "for pasture"

Line 1-4: Remove this sentence.

Line 7: This sentence is unclear.

Line 31: Because of the coarse temporal resolution of what?

Line 31: Remove this sentence.

Line 1-7: Consider moving this to Methods or removing.

Move the entire "C budget of a pasture ecosystem" to Methods

Make a separate Site section in the manuscript.

Consider moving a good portion of the Methodology to an Appendix.

Please add p-values or other values of statistical significance to the Results.

Line 11: Remove sentence beginning "This amount..."

Line 17: This occurred in both pastures?

Again, I suggest that differences in GPP and RE are actually both large, just that GPP differences > RE differences.

Line 21: Consider changing mmol C and mol H2O to grams and mm. Also, need a unit of time.

Why are the "Uncertainty analysis" and "Non-CO2" sections in Results? Please separate results from methodology in these, and put them in the correct sections of the manuscript.

Line 3: Remove "warranting the rigorous..."

Entire "C budget uncertainty" section is not appropriate for the Discussion.

Line 17: Please rewrite and clarify this sentence.

Line 21: "Efficiently used" needs to be better explained.

Line 23: Remove sentence beginning "It is instructive"

Line 32: change "C balance" to "C neutral"

Line 5: Why did this maximize GPP? Need a citation.

In Section 5.3, need to include additional sources.

Line 25: Need citations

Line 27: Explain the difference between this study and Rutledge 2015 more substantially.

Line 33: Information about management activities is important, but it is impossible to separate the influence of irrigation versus that of fertilizer in your study. It's probably best to pay this some attention, and suggest the value of better understanding the influence of these variables.

Table 3: I recommend briefly explaining why some data were not available, and why FCH4 is the same for both pastures.

Figure 1: Not a typical map, but it works.

Figure 3: Consider moving to Appendix.

Figure 8: Consider moving to Appendix.

---

## Referee Comment (RC2) · Anonymous Referee #2 · 17 Mar 2016

-Overall-

The article titled "Carbon budgets for an irrigated intensively-grazed dairy pasture and an unirrigated winter-grazed pasture" documents (1) how the annual NECB of the irrigated, intensively-managed pasture was quantified, (2) how it differed from that of an adjacent unirrigated pasture, (3) uncertainties of the annual NEP and NECB, and (4) how the management practices influenced both NEP and NECB. This research is important for assessing economic and environmental sustainability of the intensification of grazed grasslands in New Zealand. I think that this study can be an example of climate smart agriculture (http://www.fao.org/climate-smart-agriculture/72610/en/). In general, the manuscript is well written and the figures are nice presentations of the

data. The experiment was well designed to quantify the NECB of pasture reliably. The topic will likely be of interest to the readers of OzFlux special issue in BG. After careful consideration of all points raised in this review, I am favorable to recommend publication of this study in BG.

-General comments-

1. I think the ultimate goal of this study is to answer questions regarding economic and environmental sustainability of intensification of grazed grasslands through application of irrigation and fertilisers. Economic and environmental sustainability and carbon (i.e., $CO_2$ and $CH_4$) uptake/emission of pasture are directly related to the concept of climate smart agriculture. Climate-smart agriculture promotes production systems that sustainably increase productivity, resilience (adaptation), reduces/removes GHGs (mitigation), and enhances achievement of national food security and development goals (http://www.fao.org/climate-smart-agriculture/72610/en/). Please consider including this concept in the introduction, and adding that the management practices in this study site can be an example of climate smart agriculture in the conclusions. My only concern is that irrigation is economically and environmentally sustainable practice for this grazed pasture.

2. The authors selected 0.12 m s-1 of sigma_w as the low turbulence filter threshold and the NEP was quantified using 0.12 m s-1 of sigma_w threshold. The lower threshold results in the lower proportion of gaps as well as the overestimation of NEP. The results from Fig. 8 can be expected without this kind of assessment. I think that the uncertainty should be assessed using the NEP only from the higher thresholds than 0.12 m s-1 of sigma_w (i.e., the dependency of the measured nighttime CO2 flux on sigma_w was negligible). Actually, the difference between the NEPs from 0.12 m s-1 and 0.15 m s-1 of sigma_w thresholds was relatively small. In that sense, the threshold-dependent uncertainty in this study can be overestimated.

-Specifics-

[Figure]

Line 8, page 5: What is "N"?

Line 21, page 8: The process-based gap-filling method of Barr et al. (2004) is important because its results are used for estimating the uncertainties from the gap-filling procedure. Therefore, please explain more details about the method.

Line 29, page 12: How did the authors get those numbers (i.e., 22 g C m-2 yr-1 and 38 g C m-2 yr-1 of the total uncertainties). Please explain the derivation.

Line 7, page 13: Please cite a reference for '5% of annual ET.'

Line 25, page 13: What is "SE"?

Line 14-19, page 17: Please briefly compare weather/climate of the site in this study with that in Ammann et al. (2007).

---

## Referee Comment (RC3) · Anonymous Referee #3 · 21 Mar 2016

Review of "Carbon budgets for an irrigated intensively-grazed dairy pasture and an unirrigated winter-grazed pasture" published in Biogeosciences Discussions.

This study reports important numbers about net carbon budgets at two differently managed pastures and concludes that 1) net ecosystem carbon budget results show agreements with other previous studies done over grazed grassland 2) this finding is inconsistent with long-term carbon stock studies of other New Zealand pastures.

The methodologies used in this study are well described and based on the latest procedures of eddy-covariance method. Inversely, this implies that this study puts more focus on methods and numbers rather than efforts to understand underlying biogeochemical processes. Please check comments below and hope that these comments

[Figure]

help to improve this manuscript for better readability and contributions to communities.

Overall comments:

○ More extensive comparison with previous studies is needed. This study tries to compare their results with values reported by previous studies but remains to be superficial, especially to NEP. Particularly, in 5. Discussion, the authors briefly mention that studies of impacts on soil moisture and temperature on GPP and ER are needed. But I believe that this analysis can be done directly because GPP and ER are already computed with meteorological and soil variables. Do not stay in simple speculation only from NEP and go forward further with separate analysis with GPP and ER.

○ The authors wrote that annual GPP of the managed pasture was twice that of the latter but ER showed about 68

○ The authors argue that respiration by cattle is originated from carbon in the grazing term in Eq. (3), cattle respiration should be excluded from the carbon budget equation to avoid double-counting. But I am not quite sure if this argument should go to the term of excreta, if considering that carbon uptake by cattle through grazing is conserved as the sum of respiration (metabolism) and excreta while they stay in pasture. Table 3 shows that difference between excreta and grazing is about 303 and 30 gC m-2 year-1 at the managed and unmanaged pastures, respectively and I wonder how to deal with this issue properly. In addition, if considering typical values of CO2 respiration by cows per day and grazing period of about 10 days per year, these differences seems to be related to cattle respiration. This is also important in comparing this study with previous papers because previous studies did consider the cattle respiration in the carbon budget equations. More clear description is needed.

○ Several sentences are redundant in method, results and discussion sections.

Specific comments:

○ Figure 4 (b): Volumetric soil moisture sometimes exceeded 0.6 and I wonder if this large number is related to calibration issues of TDR in soils having significant clay.

○ page 6 line12: If considering 1 m tube length and 5.8 mm and 3.9 mm inner diameter, transit time of 0.36 s and 0.28 s seems to be pretty long. Please can you explain how it could be decided?

○ page 11 line 5: It seems to me that ER increased after grazing events.

○ page 12 line 26: How can we know that it is not reasonable?

○ page 13 line3-4: How can we know that it is not reasonable?

○ page 13 line 12: How can we know that it is not reasonable?

○ 5. Discussion: It will be much better if there are figures to show GPP and ER separately and extensive analysis on GPP and ER with atmospheric drivers, soil temperature, and soil moisture.

○ page 15 line 23: How can we know that it is not reasonable?

○ page 17 line 12: How can this study consider cattle respiration in comparing with other studies.

○ Figure 5. It seems to me that ecosystem respiration increased shortly after grazing events (Fig. 5). Can you explain why?

---

## Author Comment (AC2) · 7 Apr 2016

Please see the supplement PDF.

Please also note the supplement to this comment:
http://www.biogeosciences-discuss.net/bg-2016-46/bg-2016-46-AC2-supplement.pdf

―――――――――――――――――――――

---

## Author Response (AR1)

**Biogeosciences Discussions**

**Final author comment on Biogeosciences Discuss., doi:10.5194/bg-2016-46, 2016**

"Carbon budgets for an irrigated intensively-grazed dairy pasture and an unirrigated winter-grazed pasture" **by J. E. Hunt et al.**

**Response to Associate Editor Report**

Associate Editor Decision: Publish subject to minor revisions (Editor review) (18 Apr 2016) by Youngryel Ryu

Comments to the Author:

Dear Johannes

Thank you for careful revision. I see the authors logically responded to the reviewers' comments. The 1st and 3rd Reviewers concerned the scope and novelty of this manuscript, which were clarified by the authors. The 1st reviewer raised a series of editorial, terminological comments, most of them were rejected by the authors. I tend to respect the authors' argument as editorial and terminological stuff depend on style, which may be subjective.

One minor comment. I recommend clarifying how to quantify the amount of irrigation in IFR paddock (425 mm). I might be wrong but I could not find how to quantify this amount.

**Reply:** Dear Youngryel,

thank you for approving of our responses. Accordingly, we have prepared a revised version with minor changes as announced in these.

Regarding your query, we stated in the middle of Section 3.2 (P5) that "Irrigation events were identified by comparing the precipitation received at both sites." We have reformulated now to make it a bit clearer that the rain gauge at the IFR site collected total water input (precip + irrig).

The following responses to the three reviewers have, in their substance, already been published in the Interactive Discussion as individual replies. They are altered here only where it was necessary to specify which changes to the manuscript we have made in response to the reviewers' suggestions. The alterations are visible as tracked changes.

Best regards,

Johannes (on behalf of the author team).

**Response to Anonymous Referee #1**

In this manuscript, the Authors investigate the carbon (C) exchange dynamics and C balance of cattle-grazed pastures in New Zealand using the eddy covariance (EC) technique, and a technique to estimate C uptake and loss by cattle. As is the case with many grasslands and pastures globally, a mostly-ungrazed pasture was net C neutral during the one-year study period, whereas a pasture that experienced irrigation, fertilization and periodic grazing was a net C sink. Yet, the C sink dynamics of this intensively-managed pasture hinged on whether or not C uptake and loss by cattle was accounted for in the ecosystem C balance.

This manuscript is well-written, the C measurement methodology is technically-sound, and I have only nominal contextual comments. However, I am concerned that this study does not present an advancement in understanding of pasture C dynamics, nor is it up to par with other, similar studies. Specifically, I am concerned that the Authors attempt to draw conclusions on ecosystem C balance using a single year of measurements, especially because there is no true control ecosystem nor is there any pretreatment comparison. Yet, for this OZFlux special issue, I think this manuscript may ultimately be considered for publication because it focuses specifically on management issues in New Zealand. I will be happy to recommend this article for publication once my main concerns are addressed.

**Reply:** Thank you for the generally positive comments. This is the first study of irrigated pasture, which is indeed a management practice of particular relevance to New Zealand, and therefore we felt indeed that this OzFlux Special Issue would be an ideal forum to present our results. We are glad that the reviewer is in support of our judgment in this regard.

The objectives of this paper, as stated at the end of the Introduction, are to present a methodology to obtain NECB for an intensively-grazed pasture system, to carefully evaluate its uncertainty, and to identify the effects of the farm management practices. On the basis of one year's data, we did not expect to provide "advancement in understanding of pasture C dynamics". Instead, our manuscript has a strong focus on describing the methodology and ascertaining its robustness. This is because NECB was expected to be a small difference of large inputs and outputs and therefore quite sensitive to errors in these.

We are fully aware that there is no "control ecosystem" or pre-treatment comparison available. These are very difficult to both define conceptually (as different management factors always confound each other) and to find in the reality of commercial farming (where farm managers' decisions are driven by many factors but do not include suitability for, or continuity of, a research programme). We still believe that our study is valuable, in showing how and with what uncertainty the NECB of a commercial, irrigated farm operation can be obtained. We also believe that we have exercised care not to overstate the interpretation of our results.

1) The Authors need to better frame their findings in the context of existing literature on pasture/rangeland C dynamics and C balance, and also the C balance of grasslands in general. I believe that this study is best presented as a supplement to the larger body of literature that exists on this subject. It may also be useful to focus more on the weekly and seasonal dynamics of C exchanges as a way to differentiate this manuscript from longer-term studies. Potential sources include:

Felber et al. 2016. Agricultural and Forest Meteorology

McGinn et al. 2014. Journal of Environmental Quality

Oates and Jackson. 2014. Rangeland Ecology and Management

**Reply:** Thank you – these are all interesting studies covering different aspects. We have now added a mention of Oates & Jackson as useful context to our Discussion (Section 5.3). We believe that we have already presented our study as a "supplement to the larger body of literature" by discussing where our results would fit compared with the data from 21 sites collated by Rutledge et al. (2015).

The McGinn et al. study is less useful in this regard: it was at a site with very low productivity (due to the winter-cold, summer-dry climate), it does not properly account for full NECB including grazing and excreta, and it has a strong focus on enteric methane. The Felber et al. study is mainly concerned with how to make best use of GPS-tracked cow position data to obtain NEP with or without cow respiration,

not with the C dynamics of pasture. Therefore, we have decided not to include references of these two papers.

Please note that seasonal dynamics are relatively small at our site, compared to other temperate climates. The overwhelming dynamic for the IFR pasture is the repeated decimation of biomass by grazing, from grass heights of order 20 cm to about 5 cm. (This occurs on a time scale closer to monthly than weekly.) The dominant dynamic at the UUW pasture is the summer drought, which occurs in the majority of years in this region (and is a major reason for farmers to convert to irrigation). Both these dynamics are illustrated in Figs. 5 and 6 and discussed in Section 5.2.

2) Because the UUW pasture is not a true control, the Authors may wish to truncate their study period to Aug 2012 – May 2013 (so grazing did not occur in the UUW pasture during the study period). Alternatively, it may be appropriate to cite literature on the neutral C balance of other ungrazed grasslands in NZ, which would support the Authors' determination that the grassland in their study is an acceptable control.

**Reply:** We do not claim anywhere that the UUW pasture is considered a "control". It was part of the commercial farm under investigation and of some limited use to assess the tendency of what happens to the C budget of a barely-managed pasture in the same climate. We may unintentionally have caused the impression of UUW as a "control" site, with some readers, by stating near the end of the Introduction that we aimed to "determine how the annual NECB… differed…". We will have reformulated this (P3 L11-14).

We do not see any value in truncating the study period for the UUW site since that would not make it any more suitable as a "control". It can be seen in Fig. 6 that cumulative NEP from August to May was positive (ca. 80 g m$^{-2}$), but such a figure is of limited use without including what would happen over the three winter months (most likely, a reduction of cumulative NEP because respiration tends to exceed photosynthesis in winter). The winter-grazing that occurred in May was part of the management, and by reducing biomass combined with the effects of trampling, the grazing event probably amplified the net C losses from May to August 2013 that would have occurred without grazing. The observations of annual NEP being slightly positive and NECB slightly negative appear thus very reasonable and we do not understand why this information should be suppressed or truncated. In turn, we also consider it appropriate to compare the UUW site to extensively-managed pastures elsewhere, rather than to ungrazed grassland.

3) On page 16, line 15, the Authors state that the UUW pasture and IFR pasture had different grazing histories prior to the study period. Without a pretreatment comparison, I am greatly concerned that these pastures aren't comparable.

**Reply:** As stated before, we do not claim that the UUW pasture is a "control" for the IFR pasture. The two pastures were part of one farming system and served different purposes within that. The first half-paragraph of Section 5.2 (which contains the mentioned statement) serves to explain the different management aspects and interpret what their effects were.

Regarding the longer-term history, please note that before the conversion to dairying the two sites were within the same paddock. The conversion caused the same initial disturbance, cultivation, fertiliser application and seeding at both sites. Following conversion, the different managements labelled as IFR and UUW had then been applied for four years prior to our measurements.

In-text comments:

I recommend that the Authors use active voice throughout the manuscript.

**Reply:** This is a style question. The reviewer expresses his/her preference but we do not share this preference.

I recommend using negative NEE values to indicate a carbon sink or uptake by the ecosystem, and positive NEE values to indicate a carbon source or loss.

**Reply:** Again, this is the reviewer's personal preference. We considered the pros and cons of using NEE or NEP and preferred the latter because our main interest was C gains or losses of the pasture ecosystem, not of the atmosphere. NEP follows the same sign convention as NECB, while NEE has the opposite sign (Equation 1). Therefore it appeared more logical to use NEP.

Line 11: Remove sentence beginning, "Primary terms..."

**Reply:** We disagree. This sentence provides the logical connection between the previous and the following sentence.

Line 17: Differences in GPP and RE are both very large. I recommend simply stating these differences as a result.

**Reply:** We decided to give absolute numbers for GPP and RE and their relative differences. The reviewer seems to suggest giving absolute differences instead, but we do not see why that would be superior. Finding a factor 2 in GPP between the intensively-managed site and the site with little management is, in our view, quite instructive information.

Line 18: Efficiency measured as what metric or variable?

**Reply:** The details are given in the text (end of Section 4.2) and Table 1. We would have thought that "total water input" and "to produce biomass" in the queried sentence are sufficiently clear, but we have reformulated, stating now explicity that the ratio of GPP to water input is the considered variable.

Line 19: Need a stronger conclusion than this. What new information was obtained in this study?

**Reply:** This is, to our knowledge, the first study reporting NECB of irrigated pasture. It happens to be a significantly positive value, but since we have one year at one site we cannot generalise that finding. We believe the queried sentence puts our results into adequate context.

Line 24: Intensification of grasslands needs changed.

**Reply:** Agreed. We have reformulate.

Line 26: Needs citation.

**Reply:** The reviewer probably means we need to add a citation for the global intensification trend. We are happy to do that: Thornton, P. K., Livestock production: recent trends, future prospects, Phil. Trans. Roy. Soc. B 365, 2853-2867, 2010.

Line 29: Remove "for pasture"

**Reply:** Agreed, that would fit better into the second half of the sentence. We have changed accordingly.

Line 1-4: Remove this sentence.

**Reply:** We  have considered that suggestion. We have decided to change the formulation so that it is not a description of possible outcomes for NECB (which may have appeared trivial to the reviewer), but instead states that either outcome has been observed (with reference examples as before).

Line 7: This sentence is unclear.

**Reply:** OK, we  have reformulated.

Line 31: Because of the coarse temporal resolution of what?

**Reply:** Of the SOC sampling (we have inserted these words now). This is typically done several years apart, and in the meantime management practices may have changed a number of times.

Line 31: Remove this sentence.

**Reply:** We will consider that. We have decided to leave it, as a link to what follows.

Line 1-7: Consider moving this to Methods or removing.

**Reply:** This passage describes in which ways the EC methods provides valuable information to both construct annual C budgets and to interpret them (partitioning, link to ET). It is, thus, motivation for the choice of methodology and therefore serves a purpose in the Introduction.

Move the entire "C budget of a pasture ecosystem" to Methods

**Reply:** With due respect, we disagree. This section sets out the general concept of the study, not methods. Since we construct the C budget differently to most other pasture studies, by excluding the animals, it is important to have this clarified prominently and early in the paper.

Make a separate Site section in the manuscript.

**Reply:** Again, this is a matter of personal preference. It is a very common practice though to include the site description under Methods. For example, all three references cited by this reviewer in Comment 1), above, provide the site description as the first subsection under Methods. We follow the same practice here.

Consider moving a good portion of the Methodology to an Appendix.

**Reply:** We have considered that, but we have three appendices already. We provide the methodology quite detailed because we see that as a logical prerequisite for the detailed uncertainty analysis that

follows. Readers who are less interested in the details can easily skim or skip across subsections (which are clearly labelled by the three-tiered subheadings).

Please add p-values or other values of statistical significance to the Results.

**Reply:** Firstly we are not sure which of the results on this page the reviewer refers to. For the C budget terms in Section 4.2, it is clearly stated in the first sentence that the uncertainties are given in the following section. And for the environmental parameters in Section 4.1, there is hardly a need to do so, as they provide context only. Secondly, testing "statistical significance" means assessing whether an observed difference between the means of two variables has a high or low probability for having occurred randomly. Such a test requires replicated sampling of each variable. The annual C budget terms given here have no replications and thus cannot be subjected to such a test.

Line 11: Remove sentence beginning "This amount..."

**Reply:** No, this sentence provides the logical connection between the previous and the following sentence.

Line 17: This occurred in both pastures?

**Reply:** No, UUW only, and we believe that is sufficiently clear from the temporal sequence indicated by "From mid-March…" and "After that".

Again, I suggest that differences in GPP and RE are actually both large, just that GPP differences > RE differences.

**Reply:** See reply to earlier comment: we cannot see anything wrong with giving absolute numbers for GPP and RE, as well as their relative differences.

Line 21: Consider changing mmol C and mol H2O to grams and mm. Also, need a unit of time.

**Reply:** We disagree: mmol/mol is an adequate unit for water-use efficiency and is in common use. Time drops out when forming the ratio, so no time unit is required.

Why are the "Uncertainty analysis" and "Non-CO2" sections in Results? Please separate results from methodology in these, and put them in the correct sections of the manuscript.

**Reply:** The uncertainty analysis is a stated objective of the analysis and therefore an integral and important part of the Results section. The non-$CO_2$ budget terms are as important for determining NECB as the NEP term. They required their own methodologies, which are described in Sections 3.5 and 3.6 under "Methods". Section 4.4 gives the results for these terms and is thus in the "correct" place.

Line 3: Remove "warranting the rigorous..."

**Reply:** OK, this half-sentence is not essential, though it is a point for discussion.

Entire "C budget uncertainty" section is not appropriate for the Discussion.

**Reply:** We disagree. Uncertainty evaluation was a stated objective of this study, and having a good estimate of uncertainty for the C budget terms is important in order to interpret them correctly, therefore it is appropriate to discuss the uncertainty results. The contribution of the turbulence-threshold choice to NEP uncertainty deserves particular attention, as our treatment of this contains some novel aspects (Fig. 8).

Line 17: Please rewrite and clarify this sentence.

**Reply:** OK, we have done that.

Line 21: "Efficiently used" needs to be better explained.

**Reply:** The following half-sentence states precisely which ratio we consider as a measure of efficiency (GPP/WI).

Line 23: Remove sentence beginning "It is instructive"

**Reply:** The reviewer probably means to criticise that this short sentence does not convey any factual information. However, it still has a function in the text, flagging a change of focus (away from water-use, towards partitioning). Whether to retain or remove the sentence thus appears to be a question of style or taste.

Line 32: change "C balance" to "C neutral"

**Reply:** We  have reformulated this.

Line 5: Why did this maximize GPP? Need a citation.

**Reply:** To maximise GPP is the goal of the farm manager. We do not mean to say here that the observed GPP was equal to an actual (physically or biologically constrained) optimum. We have replaced "maximise GPP" with "maintain strong biomass growth throughout".

In Section 5.3, need to include additional sources.

Line 25: Need citations

**Reply:** No, the "slightly less than 200 g C m$^{-2}$ yr$^{-1}$" is not based on the literature but based on the estimation described in the preceding sentence: 50 % of grazed biomass-C from the IFR pasture equals 215 g C m$^{-2}$ yr$^{-1}$, which needs to be subtracted from the NEP excluding cows (408 g C m$^{-2}$ yr$^{-1}$) in order to obtain a redefined NEP that includes the cows' respiration. We  have inserted parentheses with these values to make the computation explicit.

Line 27: Explain the difference between this study and Rutledge 2015 more substantially.

**Reply:** The main difference regarding the C budgets is the treatment of animal respiration, precisely what is explained in the earlier parts of this paragraph. In the first sentence of this paragraph it is also stated clearly that what we are comparing to is the collection of literature data by Rutledge et al. and not just their own experiment.

Line 33: Information about management activities is important, but it is impossible to separate the influence of irrigation versus that of fertilizer in your study. It's probably best to pay this some attention, and suggest the value of better understanding the influence of these variables.

**Reply:** We agree that it is impossible to separate the effects of irrigation and fertiliser application, not only in our study but in most commercial farm settings, since both practices go hand in hand to produce more grass. One might try small-plot studies, probably best with chamber techniques, to separate out the influences of the different management factors. However, there are many potential sources for artefacts with such studies which may limit the applicability to real-world farm management. It appears more useful, for paddock-scale studies like ours, to assign observed effects to the management as a whole. For this, the management needs to be described well, and we merely point out that for the historical data in long-term C stock studies such description is not always available.

Table 3: I recommend briefly explaining why some data were not available, and why FCH4 is the same for both pastures.

**Reply:** Both recommendations concern very minor terms in the table. Regarding the first, we state in Section 4.4 that DOC leaching under irrigation was estimated based on data from a nearby experiment with a similar soil. Data without irrigation were not available. By "nd" we indicate that this term was not determined, and we do not think that an explanation why would add value to the table. Regarding the $CH_4$ emissions, the numbers are derived from the companion paper by Laubach et al. (2016). In that paper, the underlying measurements are presented, and two approaches are explored how to combine the results from two complementary micrometeorological methods. It is further explained why for $CH_4$ we opted in favour of the "merged daily" approach, which by pure happenstance led to identical annual sums for the two pastures. Had we opted for the "means combined" approach instead, or for using only one of the two methods, then the results for the two sites would have been different from each other (see Table 3 of Laubach et al.). We understand that the exact equality of the given numbers invites misinterpretations, so we will consider adding a footnote to explain this.

Figure 1: Not a typical map, but it works.

**Reply:** We  have now called it "schematic", not "map".

Figure 3: Consider moving to Appendix.

**Reply:** With all due respect, we disagree. We feel that we need to provide some justification to the reader why, for the determination of a low-turbulence threshold, we did not follow the widely-adopted practice of using u_star as the threshold variable but used sigma_w instead. This figure is part of that justification.

Figure 8: Consider moving to Appendix.

**Reply:** We considered this but have decided not to move Fig. 8. We are not aware of any published graph combining sigma_w- and u_star-thresholds in this fashion. We found it intriguing that the proportion of gaps appears to be a parameter collapsing these onto the same line, and this was not expected. The figure presents a novel finding and therefore we would like to retain it in the main body of the paper.

**Response to Anonymous Referee #2**

-Overall-

The article titled "Carbon budgets for an irrigated intensively-grazed dairy pasture and an unirrigated winter-grazed pasture" documents (1) how the annual NECB of the irrigated, intensively-managed pasture was quantified, (2) how it differed from that of an adjacent unirrigated pasture, (3) uncertainties of the annual NEP and NECB, and (4) how the management practices influenced both NEP and NECB. This research is important for assessing economic and environmental sustainability of the intensification of grazed grasslands in New Zealand. I think that this study can be an example of climate smart agriculture (http://www.fao.org/climate-smart-agriculture/72610/en/). In general, the manuscript is well written and the figures are nice presentations of the data. The experiment was well designed to quantify the NECB of pasture reliably. The topic will likely be of interest to the readers of OzFlux special issue in BG. After careful consideration of all points raised in this review, I am favorable to recommend publication of this study in BG.

**Reply:** Thank you for these endorsing comments. They show that our intentions have been understood correctly.

-General comments-

1. I think the ultimate goal of this study is to answer questions regarding economic and environmental sustainability of intensification of grazed grasslands through application of irrigation and fertilisers. Economic and environmental sustainability and carbon (i.e., CO2 and CH4) uptake/emission of pasture are directly related to the concept of climate smart agriculture. Climate-smart agriculture promotes production systems that sustainably increase productivity, resilience (adaptation), reduces/removes GHGs (mitigation), and enhances achievement of national food security and development goals (http://www.fao.org/climate-smart-agriculture/72610/en/). Please consider including this concept in the introduction, and adding that the management practices in this study site can be an example of climate smart agriculture in the conclusions. My only concern is that irrigation is economically and environmentally sustainable practice for this grazed pasture.

**Reply:** New Zealand's agricultural industry would probably be happy to brand themselves as "climate-smart" (if justified). We  have considered whether to mention this as an aspirational goal (but have decided against – in the NZ context this concept is not high on the agenda at present, and we do not wish to create a wrong impression that it was). We feel we would be overstretching the interpretation if we suggested on the basis of one year at one site that such a goal had been achieved.

2. The authors selected 0.12 m s-1 of sigma_w as the low turbulence filter threshold and the NEP was quantified using 0.12 m s-1 of sigma_w threshold. The lower threshold results in the lower proportion of gaps as well as the overestimation of NEP. The results from Fig. 8 can be expected without this kind of assessment. I think that the uncertainty should be assessed using the NEP only from the higher thresholds than 0.12 m s-1 of sigma_w (i.e., the dependency of the measured nighttime CO2 flux on sigma_w was negligible). Actually, the difference between the NEPs from 0.12 m s-1 and 0.15

m s-1 of sigma_w thresholds was relatively small. In that sense, the threshold-dependent uncertainty in this study can be overestimated.

**Reply:** With all due respect, our interpretation differs from the reviewer's. Firstly, we are not aware of any published graph combining sigma_w- and u_star-thresholds in this fashion. We found it intriguing that the proportion of gaps appears to be a parameter collapsing these onto the same line, and this was not expected. Secondly, the reviewer suggests to use the higher sigma_w threshold because the observed dependence is expected to flatten out as the threshold is increased further. That is the theory. However, our figure, including the u_star-thresholds, indicates this might not be the case in practice, and so does our Fig. 3, where the ER dependence on either threshold variable does not quite reach a "plateau" in any of the panels. Similar behaviour has been reported elsewhere, with perhaps the most striking examples in Anthoni et al. (2004), including 4 different sites. Therefore, we believe that caution is warranted and our estimate of threshold-related uncertainty should not be reduced.

-Specifics-

C2Line 8, page 5: What is "N"?

**Reply:** N = Northern. Is now spelt out in revision.

Line 21, page 8: The process-based gap-filling method of Barr et al. (2004) is important because its results are used for estimating the uncertainties from the gap-filling procedure. Therefore, please explain more details about the method.

**Reply:** Our opinion was that the reference would suffice, where the interested reader can look up the full details of the Barr et al. method. We will consider whether to add a couple of sentences stating the main features, but do not find it warranted to allocate more space than that.

Line 29, page 12: How did the authors get those numbers (i.e., 22 g C m-2 yr-1 and 38 g C m-2 yr-1 of the total uncertainties). Please explain the derivation.

**Reply:** The total uncertainties are obtained as the root of the sum of the squares of the uncertainty contributions in Table 2 (excluding footprint bias). We have added this explanation to the caption of Table 2.

Line 7, page 13: Please cite a reference for '5% of annual ET.'

**Reply:** The value of 5 % is our own estimate. It was obtained with the following considerations. The selective-sampling bias for ET is likely to be larger than for $CO_2$ flux because 1) tube attenuation for water vapour fluctuations is larger than for $CO_2$ fluctuations (hence, spectral correction more likely to be somewhat in error) and 2) since ET is positive during the day and near-zero at night, such bias not cancelling for daily means. Our estimate is compatible with the degree of closure of the energy budget (Appendix A).

Line 25, page 13: What is "SE"?

**Reply:** SE = standard error. Is now spelt out in revision.

Line 14-19, page 17: Please briefly compare weather/climate of the site in this study with that in Ammann et al. (2007).

**Reply:** Annual precipitation and mean temperature are quite similar, however at our site there is typically very little or no snow cover, unlike the Swiss site of Ammann et al. Thank you for making us aware of the similarity. We have included a statement about this in the revision.

**Response to Anonymous Referee #3**

This study reports important numbers about net carbon budgets at two differently managed pastures and concludes that 1) net ecosystem carbon budget results show agreements with other previous studies done over grazed grassland 2) this finding is inconsistent with long-term carbon stock studies of other New Zealand pastures. The methodologies used in this study are well described and based on the latest procedures of eddy-covariance method. Inversely, this implies that this study puts more focus on methods and numbers rather than efforts to understand underlying biogeochemical processes. Please check comments below and hope that these comments help to improve this manuscript for better readability and contributions to communities.

**Reply:** Thank you for the comments approving of our methodology. Indeed our manuscript has a strong focus on describing this in detail. This is because NECB was expected to be a small difference of large inputs and outputs and therefore quite sensitive to errors in these. Given that this is the first study of irrigated pasture, we felt high priority for ascertaining the robustness of our C budget results, rather than interpreting them with a view of the underlying processes.

Overall comments:

◦ More extensive comparison with previous studies is needed. This study tries to compare their results with values reported by previous studies but remains to be superficial, especially to NEP. Particularly, in 5. Discussion, the authors briefly mention that studies of impacts on soil moisture and temperature on GPP and ER are needed. But I believe that this analysis can be done directly because GPP and ER are already computed with meteorological and soil variables. Do not stay in simple speculation only from NEP and go forward further with separate analysis with GPP and ER.

**Reply:** Firstly, we are unable to find the passage that we supposedly "briefly mention". We do not state that studies of impacts of soil moisture and temperature are needed, as a large number of these already exist. For the pasture system studied here, irrigation is employed to keep the soil-moisture range small, and the other management aspects of repeated intensive grazing and fertiliser application are the strongest drivers of carbon fluxes. What we do state is that "the processes linking the distribution of soil water and nutrients with respiration need to be studied". The key word here is "distribution", which requires small-scale studies that were not undertaken here.

◦ The authors wrote that annual GPP of the managed pasture was twice that of the latter but ER showed about 68

**Reply:** this comment seems to be incomplete.

◦ The authors argue that respiration by cattle is originated from carbon in the grazing term in Eq. (3), cattle respiration should be excluded from the carbon budget equation to avoid double-counting. But I am not quite sure if this argument should go to the term of excreta, if considering that carbon uptake by cattle through grazing is conserved as the sum of respiration (metabolism) and excreta while they stay in pasture. Table 3 shows that difference between excreta and grazing is about 303 and 30 gC m-

2 year-1 at the managed and unmanaged pastures, respectively and I wonder how to deal with this issue properly. In addition, if considering typical values of CO2 respiration by cows per day and grazing period of about 10 days per year, these differences seems to be related to cattle respiration. This is also important in comparing this study with previous papers because previous studies did consider the cattle respiration in the carbon budget equations. More clear description is needed.

**Reply:** We have dealt with this "issue" properly. Two facts are crucial to understand our approach to the C budgeting of the IFR pasture. The first is that our "pasture ecosystem" is in practice one paddock, where the NEP measurements are made, biomass removal determined, grazing events recorded etc. (We assume that this paddock is representative for all paddocks of the farm that are included in the rotational-grazing operations.) The second key fact is the short duration of cattle presences: each grazing event lasted only 1 to 2 d and was followed by a cattle-free period of 20 to 30 d. Therefore, the cattle are not part of the "ecosystem" considered by us, because most of the time they are elsewhere (in other paddocks, at milking, or off-farm in winter). Regarding the C budget of the pasture ecosystem: by determining the biomass grazed we have accounted for the C exported by the cattle. The C conversions within the animals (Appendix C) are, with one exception, irrelevant to this budget because any C that is respired, converted to $CH_4$, used for milk production or kept as liveweight gain is not returned to the pasture. The exception is the excreta, which are returned to the pasture almost in their entirety (mostly by deposition during grazing, and in part as effluent collected during milking). The excreta, upon deposition, become part of the pasture ecosystem. The C returned with them must be counted as an input. From then on, any $CO_2$ emissions from the excreta are legitimately included in NEP. – The budget approach for the UUW site is identical, except that there was only one grazing event, and hence the amounts of biomass removed and excreta returned are one magnitude smaller than at the IFR site.

∘ Several sentences are redundant in method, results and discussion sections.

**Reply:** Without specific suggestions which sentences, this is difficult to reply to. We will re-read carefully. Some repetitions may have been intentional, to remind the reader of a relevant fact stated in a different section before.

Specific comments:

∘ Figure 4 (b): Volumetric soil moisture sometimes exceeded 0.6 and I wonder if this large number is related to calibration issues of TDR in soils having significant clay.

**Reply:** Soil VWC reached 0.6 once (in June, after intensive rain) but did not exceed this value. The total porosity of the soil layer from 0 to 10 cm depth was 0.66 (±0.07), as determined from three pits (S. Carrick, pers. comm.); a VWC value of 0.6 measured at 5 cm is therefore possible. (For deeper layers, the porosity was substantially smaller.) Our soil moisture sensors (Delta-T SM300) were not TDR instruments.

∘ page 6 line12: If considering 1 m tube length and 5.8 mm and 3.9 mm inner diameter, transit time of 0.36 s and 0.28 s seems to be pretty long. Please can you explain how it could be decided?

**Reply:** Thank you for querying this. This was incorrectly expressed by us. The given values represent the total time lags, which include not only the travel time along the tube but also the time to exchange the gas analyser's cell volume, of 0.016 L (half the volume of the original tube), and a fixed processing delay of 0.13 s (see LI-COR manuals). We  have  clarified this in our revision.

∘ page 11 line 5: It seems to me that ER increased after grazing events.

**Reply:** No, this was not the case. Presumably, the reviewer inferred this from Fig. 5? In this figure, grazing events are located immediately above the upper edges of dark-blue areas (sharp drop in photosynthesis due to reduction in leaf area). The effect of these events on respiration would need to be gauged from subtle colour changes during night-time hours (left and right of the blue-edge areas). If the reviewer was right, there should be transitions from lighter (orange) to darker (red). Such changes do not seem to be a general pattern, though. We illustrate this with the following figure which shows NEP across two grazing events. The events are shown as shaded bars, and the NEP data inside these are gap-filled (thus, to be ignored here). Comparing the diurnal NEP minima before and after grazing events, there is no indication in this figure that respiration increased after grazing. In general, changes in NEP minima from one day to the next appear to be mainly driven by meteorology (temperature).

[Figure]

**Figure R1.** Evolution of (a) half-hourly $CO_2$ flux density (NEP) and (b) cumulative $CO_2$ uptake over one month, including two grazing events (shaded) at the IFR paddock.

◦ page 12 line 26: How can we know that it is not reasonable?

**Reply:** Thresholds outside the ranges considered in Fig. 8 differ by 50 % or more from the thresholds obtained with MPT or CPD algorithms. As Fig. 3 illustrates, if a threshold was chosen too small then it

would fall into the range where ER steeply rises with the turbulence-indicating variable (which is the artefact one wishes to exclude). Conversely, if a threshold was chosen too large, then a large number of runs with fully-developed turbulence would be excluded and the results be biased towards certain weather patterns with stronger winds.

◦ page 13 line3-4: How can we know that it is not reasonable?

**Reply:** Presumably, the reviewer meant to ask "How can we know that this is reasonable?" (referring to 3 % uncertainty for EC and footprint)? These uncertainties are discussed for NEP in the second and third paragraph of Section 4.3. (p.11-12), and results applied here to GPP and ER in the same proportion.

◦ page 13 line 12: How can we know that it is not reasonable?

**Reply:** Presumably, the reviewer meant to ask "How can we know that this is reasonable?" (referring to the inherent robustness of neural-network gap-filling for ET)? Patterns of ET are well-studied and reasonably predictable: near-zero at night, and driven primarily by available energy and water vapour deficit, with modifications due to vegetation effects (represented in the algorithm by NDVI as a driver). The gap-filling algorithm reproduces the ET patterns in response to these drivers well.

◦ 5. Discussion: It will be much better if there are figures to show GPP and ER separately and extensive analysis on GPP and ER with atmospheric drivers, soil temperature, and soil moisture.

**Reply:** The objectives of this paper, as stated at the end of the Introduction, are to present a methodology to obtain NECB for an intensively-grazed pasture system, to carefully evaluate its uncertainty, and to identify the effects of the farm management practices. The effects of meteorological drivers have been studied many times before, and we do not expect to add new knowledge in this respect. The strongest drivers in the studied system are not the meteorological ones, but the amount of biomass and its rapid decimation with each grazing event.

◦ page 15 line 23: How can we know that it is not reasonable?

**Reply:** Presumably, the reviewer meant to ask "How can we know that this is reasonable?" (referring to "other sources of NEP uncertainty were relatively minor")? As is detailed in Section 4.3 and summarised in Table 2.

◦ page 17 line 12: How can this study consider cattle respiration in comparing with other studies.

**Reply:** This is explained in p.17 L.22-25: the cows would have respired about half of the grazed biomass-C, so by correcting for this amount we get a rough estimate of NEP for the pasture ecosystem including the cows.

◦ Figure 5. It seems to me that ecosystem respiration increased shortly after grazing events (Fig. 5). Can you explain why?

**Reply:** See reply to same question above, including Figure R1.

**Carbon budgets for an irrigated intensively-grazed dairy pasture and an unirrigated winter-grazed pasture**

John E. Hunt[1], Johannes Laubach[1], Matti Barthel[1,2], Anitra Fraser[1], and Rebecca L. Phillips[1]

[1]Landcare Research, P.O. Box 69040, Lincoln 7640, New Zealand
5   [2]Department of Environmental Systems Science, ETH Zürich, 8092 Zürich, Switzerland

*Correspondence to*: Johannes Laubach (laubachj@landcareresearch.co.nz)

**Abstract.** Intensification of pastoral agriculture is occurring rapidly across New Zealand, including increasing use of irrigation and fertiliser application in some regions. While this enables greater gross primary production (GPP) and livestock grazing intensity, the consequences for the net ecosystem carbon budget (NECB) of the pastures are poorly known. Here, we
10   determined the NECB over one year for an irrigated, fertilised, and rotationally-grazed dairy pasture and a neighbouring unirrigated, unfertilised, winter-grazed pasture. Primary terms in the NECB calculation were: net ecosystem production (NEP), biomass-carbon removed by grazing cows, and carbon (C) input from their excreta. Annual NEP was measured using the eddy-covariance method. Carbon removal was estimated with plate-meter measurements calibrated against biomass collections, pre- and post-grazing. Excreta deposition was calculated from animal feed intake. The intensively-managed
15   pasture gained C (NECB = 103 ±42 g C m$^{-2}$ yr$^{-1}$) but would have been subject to a non-significant C loss if cattle excreta had not been returned to the pasture. The unirrigated pasture was C-neutral (NECB = −13 ±23 g C m$^{-2}$ yr$^{-1}$). While annual GPP of the former was almost twice that of the latter (2679 vs. 1372 g C m$^{-2}$ yr$^{-1}$), ecosystem respiration differed by only 68 % between the two pastures (2271 vs. 1352 g C m$^{-2}$ yr$^{-1}$). The ratio of GPP to the total annual water input of the irrigated pasture was 37 % greater  than that of the unirrigated pasture, i.e. the former used the
20   water input more efficiently than the latter to produce biomass. The NECB results agree qualitatively with those from many other eddy-covariance studies of grazed grasslands, but they seem to be at odds with long-term carbon-stock studies of other New Zealand pastures.

**1 Introduction**

25   Current and predicted trends in global agriculture include that grazed grasslands are increasingly managed more intensively, through application of irrigation and fertilisers (Thornton, 2010). This aims to significantly enhance gross primary production (GPP) of pasture and thereby support more frequent rotational grazing, at higher animal densities (Tilman et al., 2001). There are questions regarding the economic and environmental sustainability of this kind of intensification, particularly if water supplies needed for irrigation become limited. Also, the effects of irrigation and fertiliser

TRIAL MODE − a valid license will remove this message. See the keywords property of this PDF for more information.

application on the carbon (C) budget are not clear. Greater GPP  may lead to greater soil organic carbon (SOC) stocks, but for pasture, the transfer of atmospheric C to the SOC pool is largely dependent on grazing and irrigation management decisions (Ammann et al., 2007; Merbold et al., 2014).  For some pastures, gains in GPP were offset by ecosystem respiration (ER) and C export  (Ammann et al., 2007; Kelliher et al., 2012), so that a net loss

5 of C and a negative net ecosystem carbon balance (NECB)  result. For others, C inputs  exceed losses, leading to a positive NECB (Ammann et al., 2007; Merbold et al., 2014).

[revised manuscript text omitted]

**5.2 Effects of management practices**

The two pastures used in this study were managed differently with respect to irrigation, fertiliser application and grazing (Fig. 2), and so while differences in NEP and NECB may be in response to management overall, it is not possible to tease apart the effect of its individual components. Until mid-December, management made no difference on cumulative ET (Fig. 7), and little on cumulative NEP. By this point in time, the UUW pasture production (initially suffering from the impact of a winter-grazing that had occurred before measurements started) had caught up with the IFR pasture, both reaching about 290 g C m$^{-2}$ (Fig. 6). From mid-December onwards, the UUW pasture turned from a C sink into a C source, as a consequence of the depleted soil water under the UUW pasturecontent (Fig. 4b), began to impact strongly, turning this pasture from a C sink into a C source, while the intensively-managed IFR pasture remained fully productive throughout summer and evaporated at higher rates than the UUW pasture. Clearly, the main effect of IFR-pasture management was to counteract water stress as a principle controlling factor in NEP. Further, the additional water input from irrigation was efficiently used, since the ratio of GPP to total water input for the IFR pasture (2.79 mmol C mol$^{-1}$ H$_2$O) was significantly greater than for the UUW pasture (2.03 mmol C mol$^{-1}$ H$_2$O), by 37 % (Table 1).

It is instructive to compare the partitioning of NEP. For the IFR pasture, GPP was 95 % greater than for the UUW pasture, while ER was only 68 % greater (Table 1). Thus, additional C uptake outpacing additional respiration explains the stronger

TRIAL MODE − a valid license will remove this message. See the keywords property of this PDF for more information.

C sink at the IFR pasture. From a management perspective, the additional water and fertiliser inputs were highly efficient, by allowing to nearly-double the production of biomass, but only a third of the additional C removed from the atmosphere was converted into feed intake for the cows and two-thirds were respired. In part, the more muted response of ER to management (compared to GPP) may be linked to lower soil temperatures during summer (Fig. 4a), as an indirect effect of irrigation (Rajan et al., 2013). To fully understand the fate of C in managed pasture systems, the processes linking the distribution of soil water and nutrients with respiration need to be studied (Conant et al., 2001; Soussana et al., 2007b; Rutledge et al., 2015).

[revised manuscript text omitted]

TRIAL MODE − a valid license will remove this message. See the keywords property of this PDF for more information.

The effect of the footprint fractions on the $CO_2$ fluxes was assessed as follows. For each run with valid fluxes, the fluxes at the two sites were linearly combined, weighted by their respective footprint fractions, to give $CO_2$ emission/uptake rates for the two underlying pastures (UUW and IFR). This procedure was equivalent to the matrix approach of Mukherjee et al. (2015). At each site, the uncorrected nighttime and daytime fluxes were summed separately, and so were their footprint-corrected counterparts. The differences between corrected and uncorrected sums gave the nighttime and daytime footprint biases. At the UUW site, both nocturnal $CO_2$ emission and daytime $CO_2$ net uptake in winter were overestimated without footprint correction. Since nocturnal $CO_2$ emissions were overestimated by a larger fraction than daytime $CO_2$ net uptake, the annual NEP of the UUW pasture was slightly underestimated, by 0.7 g C m$^{-2}$ yr$^{-1}$. At the IFR site, the biases resulting from the UUW footprint contributions were 12.1 and −4.8 g C m$^{-2}$ yr$^{-1}$ for nocturnal $CO_2$ emission and daytime $CO_2$ net uptake, respectively. This means that without footprint correction, both processes were underestimated, but the nocturnal respiration bias was larger, and hence NEP was overestimated by 7.3 g C m$^{-2}$ yr$^{-1}$. This may appear surprising, but at night the source area extends farther than during the day, and the winter season contributed most to the footprint bias, when the photosynthetic uptake rates were smallest.

**Appendix C. Carbon inputs and outputs of the dairy herd**

In the main text, we report the measurements of the biomass removed by grazing, per area, and its C content. This appendix serves the following purposes: the per-area estimates are related to estimates per cow and per day, it is described how C deposition with excreta is linked to the biomass intake (from pasture and supplements combined), the C content of the milk produced is calculated, and it is assessed whether all these estimates are consistent and realistic. A full C budget of the dairy herd is not constructed, as that would require independent estimates of the cows' $CO_2$ respiration, $CH_4$ emissions, and liveweight gains, which were not obtained here.

**C1. Seasonal budget and area considerations**

The milking season was 248 d long (25 Sept 2012 to 29 May 2013). There were 868 cows, managed in two herds. The total irrigated-pasture area of the farm was 328 ha. Dividing this area by the previous two numbers gives a factor, $S$, which converts from amounts per grazing area per year to amounts per cow per day. This factor was thus $S = 15.25$ m$^2$ yr cow$^{-1}$ d$^{-1}$.

**C2. Biomass removed by grazing**

The biomass dry-matter (DM) removed by grazing from the IFR pasture was determined as 10,500 (±234) kg DM ha$^{-1}$ yr$^{-1}$, and the C contained therein as 430 (±14) g C m$^{-2}$ yr$^{-1}$ (Sect. 4.4 of main text). Using the conversion factor $S$, each cow consumed 16.0 (±0.35) kg DM cow$^{-1}$ d$^{-1}$ from the IFR pasture, containing 6.56 (±0.21) kg C cow$^{-1}$ d$^{-1}$.

TRIAL MODE − a valid license will remove this message. See the keywords property of this PDF for more information.

**C3. Supplementary feed and total feed intake**

[revised manuscript text omitted]

TRIAL MODE − a valid license will remove this message. See the keywords property of this PDF for more information.

**Figures**

[Figure]

5 **Figure 1.**  **Schematic of paddocks and instrument layout on a commercial dairy farm on the Canterbury Plains of New Zealand. Areas of irrigated, fertilised and rotationally grazed pasture (IFR) are shaded in grey. An area of unirrigated, unfertilised and winter-grazed pasture (UUW) is wedged between two pivot-irrigation circles. The measurement sites in the IFR and UUW paddocks are indicated by mast symbols. An instrument hut was situated on the boundary between the two paddocks (filled square).**

TRIAL MODE − a valid license will remove this message. See the keywords property of this PDF for more information.

[Figure]

**Figure 2. Pasture management regime from 17 August 2012 to 16 August 2013. Bars indicate the time and duration of grazing events in the UUW and IFR paddock. Crosses and vertical lines mark times of fertiliser and irrigation applications, respectively, to**
5  **the IFR paddock.**

TRIAL MODE − a valid license will remove this message. See the keywords property of this PDF for more information.

[Figure]

Figure 3. Example of moving-point threshold detection for night-time CO$_2$ fluxes (**ER**), for the temperature class 13 to 16 °C (soil temperature at 20 mm depth), at the UUW site (top) and the IFR site (bottom). In Panels a and c the standard deviation of vertical wind speed ($\sigma_w$) is used as the discriminating variable, following Acevedo et al. (2009); in Panels b and d, friction velocity (**u\***) is used instead, as is common practice. Bin means and their standard errors are indicated by dots with error bars. The solid line marks 0.99 times the mean of all bin-means to the right of the actual bin. The vertical dashed lines indicate the overall optimum threshold choices for $\sigma_w$ and **u\***, respectively.

TRIAL MODE − a valid license will remove this message. See the keywords property of this PDF for more information.

[Figure]

**Figure 4. Time series at the UUW and IFR sites for daily means of (a) soil temperature and (b) soil moisture at 50 mm depth; (c) daily sums of rainfall (both sites) and irrigation (IFR only); and (d) daily values of the Normalised Difference Vegetation Index (NDVI).**

TRIAL MODE − a valid license will remove this message. See the keywords property of this PDF for more information.

[Figure]

**Figure 5. Visualisation of diurnal variation (horizontally) and seasonal variation (vertically) of net ecosystem production (g C m$^{-2}$ h$^{-1}$) for the UUW and IFR pastures. Positive values (in yellow, green and blue) represent net C uptake by the ecosystem, and negative values (in orange and red) represent net C losses.**

TRIAL MODE − a valid license will remove this message. See the keywords property of this PDF for more information.

[Figure]

**Figure 6. Cumulative net ecosystem production for the irrigated, fertilised and rotationally grazed (IFR) and the unirrigated, unfertilised and winter-grazed (UUW) pasture. Positive values represent C uptake by the ecosystem.**

TRIAL MODE − a valid license will remove this message. See the keywords property of this PDF for more information.

[Figure]

**Figure 7. Cumulative water input (sum of rainfall and irrigation) and evapotranspiration (ET) for the IFR and UUW pasture.**

TRIAL MODE − a valid license will remove this message. See the keywords property of this PDF for more information.

[Figure]

**Figure 8. Variation of annual NEP estimate for the IFR pasture with the fraction of data gaps, as resulting from the choice of low-turbulence filter threshold. The $\sigma_w$ thresholds (dots) are, from left to right, 0.08, 0.10, 0.12 and 0.15 m s$^{-1}$; the $u_*$ thresholds (triangles) are 0.12, 0.15 and 0.18 m s$^{-1}$. The optimal threshold, based on moving-point threshold detection, is indicated by a circle. The solid line, obtained by linear regression, is displayed only to indicate the qualitative trend.**

TRIAL MODE − a valid license will remove this message. See the keywords property of this PDF for more information.